# Precision cutaneous stimulation in freely moving mice

**Isobel Parkes, Ara Schorscher-Petcu, Qinyi Gan, Liam E Browne***

Wolfson Institute for Biomedical Research, University College London, London, United Kingdom

## eLife Assessment

This **important** study combines real-time key point tracking with transdermal activation of sensory neurons as a general technique to explore how somatosensory stimulation impacts behavior in freely moving mice. After addressing concerns about classification of the behavioral responses to nociceptor stimulation, the authors now **convincingly** demonstrate a state-dependence in the behavioral response following nociceptor activation, highlighting how their real-time optogenetic stimulation capabilities can yield new insights into complex sensory processing. This work is a technological advancement that will be of interest to a broad readership, in particular labs studying somatosensation, enabling rigorous investigation of behaviors that were previously difficult or impossible to study.

***For correspondence:**
liam.browne@ucl.ac.uk

**Competing interest:** The authors declare that no competing interests exist.

**Abstract** Somatosensation connects animals to their immediate environment, shaping critical behaviors essential for adaptation, learning, and survival. Probing the relationships between somatosensory inputs and behavior in mice presents substantial challenges, primarily due to the practical difficulties of delivering stimuli to the skin during movement. To address this problem, we have developed a system for precise cutaneous stimulation of mice as they walk and run through environments. The system employs real-time body part tracking and targeted optical stimuli, offering precision while preserving the naturalistic context of the behaviors studied, thereby overcoming the traditional trade-offs between precision and animal behavior. We demonstrate the system from nociceptive testing conducted in standard small chambers to behavior in large complex environments, such as mazes. We observed that cutaneous inputs evoke rapid responses, which modify behavior when stimuli are applied during motion. This system provides a means to explore the diverse and integrative nature of somatosensation, from reflexes to decision-making, in naturalistic settings.

## Introduction

Survival and adaptation are supported by the somatosensory system, which provides signals from the body surface to drive and refine behavior. It links the skin and central nervous system, recruiting reflexes and shaping behavior through learning and fine-tuning responses. For instance, a noxious stimulus at the skin should generate appropriate responses that minimize harm and increase the odds of survival. However, probing the somatosensory system in freely moving animals presents significant challenges due to the trade-offs between precision and preserving natural behavior.

Current methods for cutaneous stimulation usually necessitate direct physical contact, which restricts the range of behaviors and environments that can be studied. These methods often involve restraining animals or confining them to small chambers where stimuli are applied to their body. While valuable for understanding immediate responses, such approaches fail to capture the complexity of naturalistic behaviors. There is a growing need to study freely moving mice in dynamic, naturalistic

environments (*Juavinett et al., 2019*; *Datta et al., 2019*; *Dennis et al., 2021*; *Rosenberg et al., 2021*; *Zong et al., 2022*), moving away from restraining and restricting behavior. However, delivering precise cutaneous stimuli to mice navigating complex settings, such as mazes, remains a significant challenge.

Experiments can proceed in more complex environments without researchers in proximity when cutaneous stimuli, such as electric shocks are delivered via a grid floor. This approach has contributed significantly to our understanding over decades, being used in learning studies to probe cells and circuits underpinning aversion, avoidance, fear, expectancy, and memory formation and retrieval. However, the underlying somatosensory processes cannot be resolved due to a lack of spatial precision and control—they indiscriminately stimulate multiple body parts in contact with the floor.

There have been attempts to bridge these gaps by applying localized cutaneous stimuli (e.g. von Frey filaments) to moving rodents. This has been shown to be useful in recent studies of circuits involved in pain processing (*Cheng et al., 2017*; *Gangadharan et al., 2022*). However, these approaches require experimenters to be in close proximity to the animals, to continuously observe their movements as they explore, and to manually touch a hind paw at regular intervals (*LaBuda and Fuchs, 2000*; *Uhelski et al., 2012*). This can cause observer bias and limit both precision and the complexity of the environments.

To address these challenges, we developed a closed-loop system to automatically deliver spatio-temporally precise cutaneous stimuli in a remote and dynamic manner, targeting mice as they freely explored environments. The system leverages advances in remote transdermal optogenetic stimulation (*Schorscher-Petcu et al., 2021*) and real-time markerless body part (keypoint) tracking (*Kane et al., 2020*). The approach moves away from restricted and restrained behavior to enable behaviorally relevant stimulation in more naturalistic environments (*Dennis et al., 2021*). This addresses the traditional trade-offs between precision and naturalistic environmental settings, providing a framework in which to integrate reflex recruitment, motivation, learning, and decision-making.

## Results

To develop closed-loop cutaneous stimulation in mice exploring large environments, we built a system that could rapidly track and target mice and stimulate them remotely (*Figure 1*). We used real-time pose estimation to monitor body parts and target lasers for spatiotemporally precise thermal stimulation and optogenetic stimulation of genetically defined nociceptor fibers at the skin surface

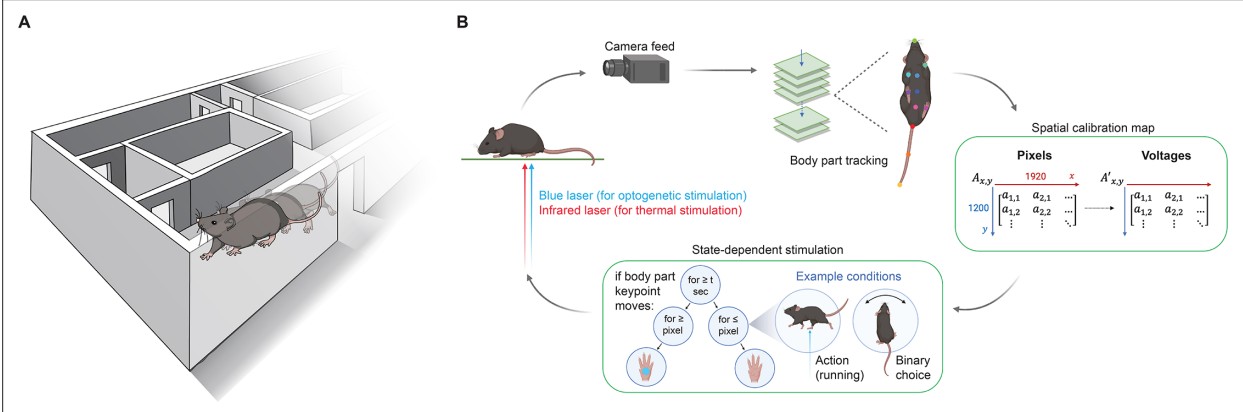

**Figure 1.** Closed-loop cutaneous stimulation of mice freely moving in naturalistic environments. (**A**) Mice can be remotely targeted with cutaneous stimuli while freely exploring complex environments, such as a maze. (**B**) Schematic illustrating the closed-loop control workflow. A freely moving mouse is recorded using a camera feed, enabling real-time pose estimation to track multiple body part keypoints. The extracted frame keypoint (x, y) of a selected body part is converted to pre-mapped *x, y* mirror galvanometer control signals to steer the laser beam paths and pulse the lasers. The movement of the galvanometer mirrors and triggering of the laser are determined by pre-programmed behavioral or environmental conditions, allowing stimulation to depend on behaviorally relevant states: for example, if the mouse was performing specific actions (running, sleeping, grooming, rearing) or making choices (turning right in a maze, exploring a specific area of the environment). Flexible, state-dependent laser targeting was accomplished using an infrared laser for thermal stimulation and a blue laser for optogenetic stimulation of genetically targeted primary afferent neurons, enabling high spatiotemporal control of stimulation to small areas of skin. Schematics in panel B were created using BioRender.com.

(*Schorscher-Petcu et al., 2021*; *Kane et al., 2020*). The approach was demonstrated in large environments, allowing automated cutaneous stimulation in an arena and during goal-directed behavior as mice run through a maze.

## A system for closed-loop cutaneous stimulation

We designed a system that automatically tracks targets for cutaneous stimuli across a large environment, leveraging our laser-mirror galvanometer design (*Schorscher-Petcu et al., 2021*). The environment was built on a large glass platform, allowing us to record exploring mice from below with a camera (*Figure 2A and B*). The camera enabled real-time tracking with DeepLabCut-Live! (*Kane et al., 2020*), estimating the keypoints of multiple body parts for every frame of the camera feed. The target body part keypoint *x* and *y* coordinates were converted to pre-mapped control signals for *x* and *y* galvanometer mirrors to steer lasers as required. This provides a closed-loop system for dynamic control of stimuli according to behaviorally relevant criteria.

The system was optically precise. The glass platform was close to 1 m above the galvanometers, resulting in a maximum focal length variability of 3.49%, minimizing differences in the size of the laser spot in the stimulation plane. The absolute optical power and power density were uniform across the glass platform (coefficient of variation 0.035 and 0.029, respectively; *Figure 2—figure supplement 1A*). Laser power could be modulated using the analog control. The laser spot size was set to $2.00 \pm 0.08$ mm$^2$ (mean ± SD; coefficient of variation = 0.039) at the stimulation plane with a series of lenses along the blue light beam path (*Schorscher-Petcu et al., 2021*). The laser spot could be moved along specific trajectories creating patterns (*Figure 2C*, *Figure 2—figure supplement 1B*). We used 10,000 *x* and *y* voltage pairs to jump the laser across the stimulation plane and map the voltages to corresponding pixels (*Figure 2C*). Surface fits resulted in a pixel-voltage mapping dictionary that minimized non-linear distortions. This resulted in a mean average Euclidean error (MAE) of 1.2 pixels (0.54 mm) between predicted and actual laser spot locations in the 500 mm arena. The system and glass platform were stable, showing a displacement equivalent to about half the width of a hind paw (MAE = $1.94 \pm 2.61$ mm) each week during intensive use of the system (*Figure 2—figure supplement 1C*), which can be corrected in less than 30 min by remapping.

The system could accurately target moving mice. Real-time estimation of keypoints (*Figure 2D and E*) was used for closed-loop control of the galvanometer mirror angles, resulting in pairwise correlations of *r*=0.999 between galvanometer coordinates and hind paw keypoints (*Figure 2F*). Thus, the laser could be targeted in real time to body parts when certain programmatic criteria were met (see *Figure 2—figure supplement 2* for the information flow). For instance, the laser beam could be triggered if the distance of an individual keypoint moves with variance ≤*v* for time ≥*t* and keypoint estimation likelihood was ≥*l*, where *v*, *t*, and *l* are user-defined variables. To determine the targeting accuracy, we used wild-type mice that did not express ChR2 so that blue light pulses did not cause behavioral responses. The latency between acquiring a frame 1100×1100 pixels, estimating keypoints, and targeting a laser was $84 \pm 12$ ms (mean ± SD using 16,000 trials across four wild-type mice; *Figure 2—figure supplement 1D*). This delay is sufficient to target paws: during locomotion, the hind paws were static for $350 \pm 44$ ms in the stance phase and moving for $100 \pm 1$ ms in the swing phase (*Figure 2—figure supplement 1E*). The positioning of paws during the stance phase of locomotion creates 'footprints' in keypoint space, indicating moments when the paws are momentarily static even as the mouse moves (*Figure 2G*).

Mice move around at variable speeds while exploring, which can be categorized (*Figure 2H*). During these movements, locomotor gait is defined by the coordinated sequence and timing of stance and swing phases across all four limbs. Each paw alternates between the stance and swing phase; thus, as the mouse moves forward, the individual paws are stationary during their respective stance phases. When the mouse was stationary ($58 \pm 7$% of the time), the hind paws were static in $99.8 \pm 0.1$% frames, and this remains high even as the mouse increases its bodily speed: $95.7 \pm 0.1$% frames for 'low' speed ($28 \pm 4$% of the time), $79.3 \pm 0.1$% frames at 'medium' speed ($8 \pm 2$% of the time), and $60 \pm 0.3$% at 'high' speed ($6 \pm 1$% of the time). Therefore, even during locomotion, the hind paws are static long enough for stimulation with short-latency body part tracking (*Figure 2I and J*). The laser was successfully targeted to the hind paws with a high 'hit accuracy' when the mouse was moving at low speeds ($95.5 \pm 2.6$%; *Figure 2K*). Hit accuracy refers to the percentage of trials in which the laser successfully targeted ('hit') the intended hind paw. As expected, the hit accuracy was reduced during high-speed

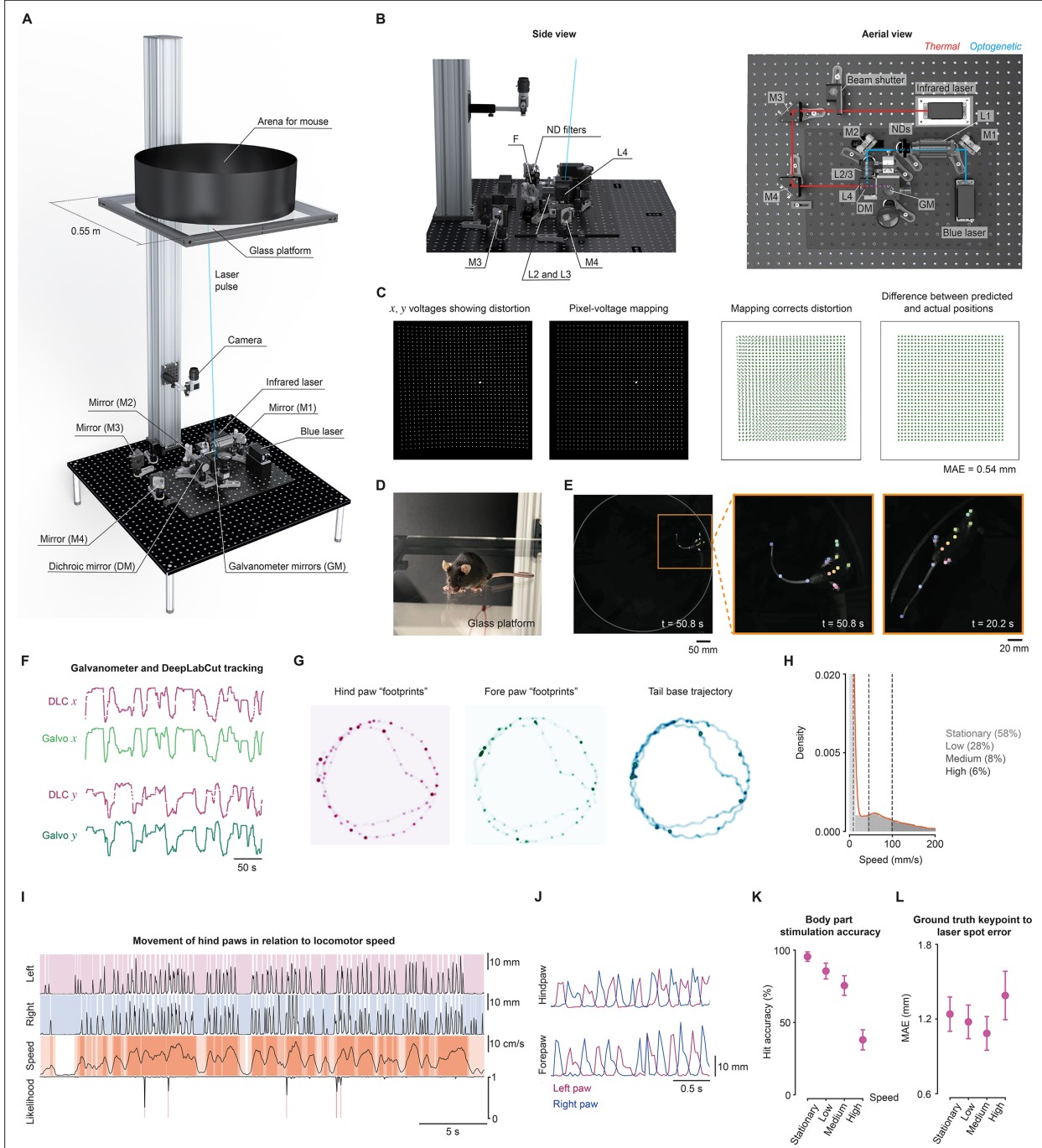

**Figure 2.** A system for closed-loop cutaneous stimulation. (**A**) Rendering of the system shows camera and stimulation optics 1 m below the glass platform, accommodating a large circular arena (0.5 m diameter) for freely moving mice. (**B**) Side and aerial views. The blue laser beam (*blue*) was aligned to the galvanometer mirrors (*GM*) using mirrors (M1, M2), and lenses (L1, L2, L3) via ND filters. The infrared laser beam (*red*) is directed through a beam shutter with mirrors (M3, M4) and lens (L4). Converged beams in *purple*. (**C**) Average image of the laser across a linear voltage grid (*left*) and a pixel grid after mapping (*left-middle*). Pixel-voltage mapping corrects distortion (*right-middle* to *right*). (**D**) A mouse on the platform. (**E**) Tracking in the arena. (**F**) Galvanometer mirrors tracking the left hind paw keypoint. (**G**) 2D histograms of paw keypoints highlight the dwell of the locomotor stance phase compared to tail base motion. (**H**) Histogram of tail base speed indicating categories from four wild-type mice (16,000 frames). (**I**) Keypoint traces illustrating the out-of-phase swing-stance during locomotion. The left hind paw trace is shown in *pink*, while the right hind paw trace is shown in *blue*. The tail base speed is shown in *orange*. (**J**) Traces showing alternating left and right paw movement. (**K**) Accuracy of the laser targeting the hind paws across speed categories. (**L**) Error between the ground truth keypoint and laser spot in these same four mice, expressed as mean average Euclidean error (MAE). See also related *Figure 2—figure supplements 1 and 2*. Renderings were created using Solidworks.

*Figure 2 continued on next page*

*Figure 2 continued*

The online version of this article includes the following figure supplement(s) for figure 2:

**Figure supplement 1.** Spatial and temporal characterization of the closed-loop optical system.

**Figure supplement 2.** Hardware and software information flow design.

running. We found zero keypoint confusion across all speeds, with 657 out of 657 trials successfully targeting the correct hind paw. The optical system targeted the hind paws of wild-type mice exploring an open arena 650±30 times within 5 min at 30 frames per second (fps, n=4 mice), providing multiple stimuli in short periods of time. We also targeted the forepaws, but confusion between them resulted in 5.7±2.2% targets being directed to the incorrect paw. This resulted in a low hit accuracy compared to the hind paws (*Figure 2—figure supplement 1F*). The laser spots were delivered with high accuracy to the targeted body parts, showing a small error of ≈1.3 mm MAE (*Figure 2L*). As laser targeting relies on real-time tracking to direct the laser to the specified body part, this metric includes any errors introduced by tracking and targeting. Thus, this design resulted in a fully automated system that facilitates spatiotemporally precise optical targeting of freely moving mice in a large environment.

## Cutaneous stimulation in large environments drives behavioral responses

We next used the closed-loop system to automatically deliver optogenetic cutaneous stimuli and examine the resultant behavior. A large circular arena was chosen to encourage free exploration and movement (*Figure 3A and B*), and behavior was examined by pose estimation, reconstructing the movement trajectories of body parts in the arena (*Figure 3C*). Brief transdermal optogenetic stimulation of nociceptors was used to achieve minimal cutaneous stimulation: mice expressed the blue-light-sensitive opsin, ChR2, in nociceptors innervating the skin (Trpv1::ChR2; *Schorscher-Petcu et al., 2021*; *Browne et al., 2017*; *Black et al., 2020*). Mice explored the arena, and when stationary were stimulated on the hind paw with brief 10 ms pulses of light, delivered at intervals of at least 10 mins (*Figure 3B*). We found that the system accurately targeted the laser to the hind paw (*Figure 3D*).

The precise cutaneous input gave rise to motor output as coordinated behavior in freely moving mice. The brief stimuli caused paw withdrawals along with coordinated whole-body behaviors, such as rapid head orientation and body repositioning (*Figure 3E and F*). These behaviors were quantified as time-locked keypoint traces for each body part (*Figure 3E and F*). Littermate control mice that do not express ChR2 did not exhibit responses to optogenetic cutaneous stimuli, including paw withdrawals or whole-body behaviors (n=10 mice). Therefore, the system enables fully automated and precise optical delivery of cutaneous stimuli to freely moving mice while simultaneously recording and quantifying behavior.

## Multi-animal stimulation for automatic nociceptive testing

The system could deliver cutaneous stimuli across a large space, evoking behavioral responses. Next, we demonstrate the flexibility of the system design and establish automatic nociceptive testing across multiple mice simultaneously. A method for random-access targeting was developed (*Figure 4A*) to target nine mice (3×3 configuration) in individual chambers; we detected idle mice by analyzing motion energy in each chamber, then rapidly selected and cropped to one chamber for real-time pose estimation and stimulation (*Figure 4B*). This reduced the computational burden by decreasing image resolution, compared to running real-time pose estimation across the whole environment (*Kane et al., 2020*). The process operated as a loop, ensuring that automated stimuli were spaced by at least 1 min apart for each mouse.

To test the system, we used: (1) thermal stimulation with a 10 s pulse of infrared light (785 nm) on the hind paw of wild-type mice; and (2) optogenetic stimulation of cutaneous nociceptors with a 3 ms pulse of blue light (473 nm) on the hind paw of Trpv1::ChR2 mice. We varied the intensity of the optogenetic stimuli using 10 Hz pulse trains (0.5–8 mW/mm$^2$) compared to a single pulse at higher intensity (40 mW/mm$^2$).

Thermal and optogenetic stimulation induced similar nocifensive behaviors, including paw responses and whole-body movements (*Figure 4C*). This can be seen from traces of hind paw movement following thermal or optogenetic stimuli (*Figure 4D–F*). Consistent with previous studies, we

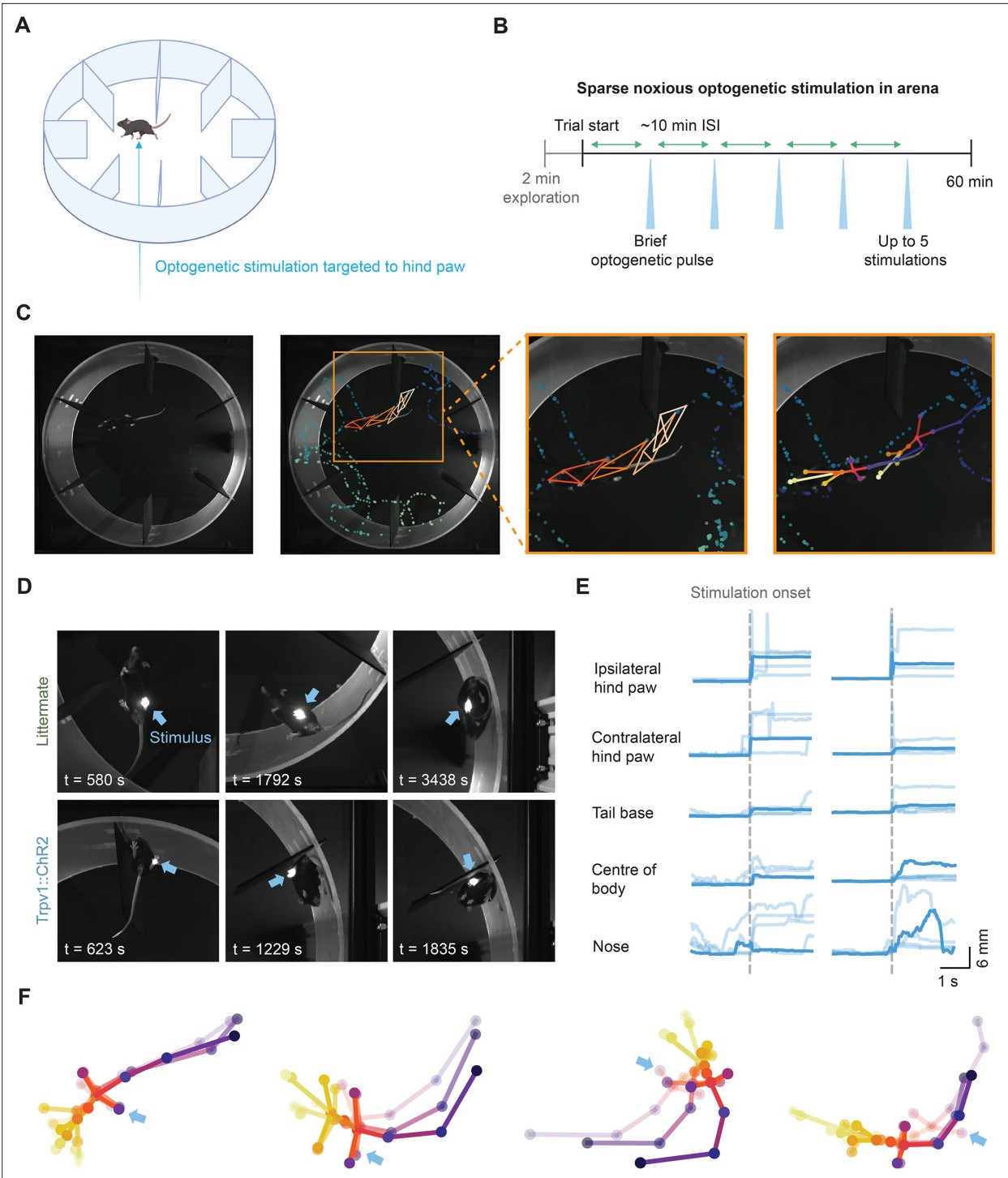

**Figure 3.** Cutaneous stimulation in large environments drives behavioral responses. (**A**) Schematic of the open arena. (**B**) Protocol for minimal cutaneous stimulation using transdermal optogenetic activation of cutaneous nociceptors (Trpv1::ChR2, n=10 mice). (**C**) A single frame showing a mouse exploring the open arena (*left*). Keypoints for the left hind paw for 750 frames prior to and 1750 frames after the frame (1 min 23.33 s duration, *middle*). The body and head orientation at four time points are shown as *orange* rhombi connecting snout, left, and right forepaw, and tail base (*middle*). Keypoint skeletons (*right*). (**D**) Representative images of a 10 ms laser pulse spot targeting the plantar surface of the hind paw in littermate (*top*) and Trpv1::ChR2 (*bottom*) mice. (**E**) Representative keypoint traces during stimulation of the left hind paw for two mice (columns): five trials with one trial shown in bold across body parts (rows) for each mouse. (**F**) Example keypoint skeletons from Trpv1::ChR2 mice showing orienting behavior to hind paw stimulation (indicated by the *blue* arrow). Schematics in panel A were created using BioRender.com.

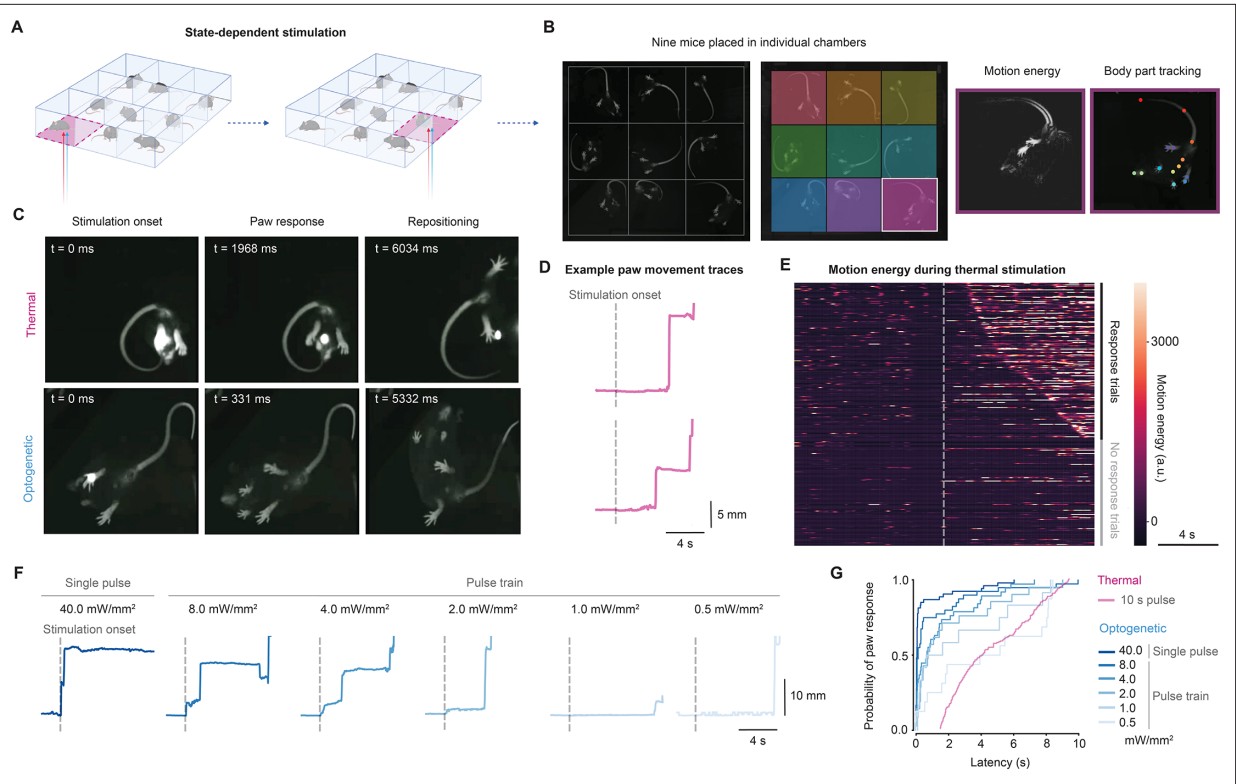

**Figure 4.** Multi-animal stimulation for automatic nociceptive testing. (**A**) Concept of the random-access multi-animal stimulation. Motion energy was used to detect idle mice in multiple chambers, randomly selecting and cropping to one chamber for real-time pose estimation and stimulation. A laser spot was targeted to the hind paw of the mouse placed in the chamber. The process looped through each of the chambers, automatically targeting and stimulating the mice. (**B**) An example camera frame highlighting the chambers with different colors (*left*). Motion energy and body part keypoints shown for an individual chamber (*right*). (**C**) Representative paw responses and body repositioning following thermal stimulation (10 s pulse) and optogenetic stimulation (3 ms pulse). (**D**) Representative paw responses during thermal stimulation of wild-type mice. Two traces plotted with keypoints are shown. The *gray dashed* line indicates laser stimulation onset. (**E**) Raster plot of motion energy during thermal stimulation trials for 18 wild-type mice (315 trials), sorted by response latency. (**F**) Representative hind paw responses during optogenetic stimulation of Trpv1::ChR2 mice. The *gray dashed* line indicates stimulation onset. (**G**) Cumulative distribution of paw response latencies to thermal and optogenetic stimulation. Thermal: 18 wild-type mice, 315 trials. Optogenetic: 9 Trpv1::ChR2 mice, 181 trials (range of 15–24 trials for individual mice). Schematics in panel A were created using BioRender.com.

observed the whole-body behaviors, such as head orienting concurrent with local withdrawal (*Browne et al., 2017*; *Blivis et al., 2017*). In the case of thermal stimulation, we analyzed the local motion energy of the stimulated hind paw movements to quantify the response. Following stimulation onset, mice showed a mean response latency of 4.74±2.48 s (mean ± SD, 188 response trials; *Figure 4E*). Cumulative distributions further demonstrated nocifensive behaviors in response to infrared stimulation (*Figure 4G*) and illustrated the temporal resolution afforded by transdermal optogenetics compared to infrared stimulation. Optogenetic stimulation-induced response latencies followed the rank order of stimulus intensity, demonstrating the dynamic range possible with this system (0.5 mW/mm²: 2.98±3.19 s, 1.0 mW/mm²: 2.84±3.19 s, 2.0 mW/mm²: 1.69±2.14 s, 4.0 mW/mm²: 1.72±2.25 s, 8.0 mW/mm²: 1.51±2.19 s, 40.0 mW/mm²: 0.81±1.65 s, mean ± SD; *Figure 4G*). In contrast, littermate control mice did not exhibit responses (*Schorscher-Petcu et al., 2021*; *Browne et al., 2017*).

This demonstrates that the system is both versatile and precise, enabling the study of local paw withdrawals and whole-body movements, including head orientation and body repositioning. The platform is large enough to accommodate 25 mice (5×5 configuration) in a single session, allowing for high-throughput automated experiments.

## Closed-loop cutaneous stimulation in mice running through a maze

We demonstrate that freely moving mice can be stimulated as they were running through a maze environment. The ability to deliver cutaneous stimuli to mice moving through behaviorally relevant tasks has previously been a significant challenge, and experimenters typically apply stimuli to moving

mice manually (*Cheng et al., 2017*; *Gangadharan et al., 2022*; *LaBuda and Fuchs, 2000*; *Uhelski et al., 2012*).

To motivate movement in a task, we built a novel maze that encourages alternation between two rewards at separate locations. The maze had one-way doors, ensuring that after making a decision to turn left or right, mice could obtain a reward but are required to circle back to the maze's start point to re-initiate the action-reward cycle (*Figure 5A and B*). The reward ports were activated by a brief nose poke and delivered a drop of sucrose water, followed by a timeout period. In addition to the timeout, the reward ports were only reset once the mouse exited the reward chamber through a one-way door, as determined by real-time keypoint tracking. The combination of a long timeout and required exit from the reward chamber rendered it more time-efficient to cycle between the two reward chambers and encouraged running along the corridors.

Mice were trained over three sessions in the maze. In the first training session, they rapidly navigated the one-way doors with little delay after a few attempts. This quickly led to the use of reward ports in the chambers on either side of the maze. The first reward was collected after 440±117 s (4 mice). By the third training session, this time decreased to 82±38 s, suggesting rapid learning. During the reward port timeout, mice typically explored the corridor connected to the reward chamber or exited it to explore the entry corridor. By the third day of training, the number of completed trials had increased for all mice. On average, mice completed 64±24 rewarded trials in the third training session (<2 hr in duration), although individual performance varied considerably. For example, one mouse completed only 10 rewarded trials in the third session, and another mouse completed 97 trials (*Figure 5C*).

Mice could be stimulated while they were running. Example movement trajectories from one mouse are shown in *Figure 5D*. Using Trpv1::ChR2 mice, we demonstrated the system's utility in studying localized cutaneous hypersensitivity. We employed a widely used model of inflammatory pain with a unilateral injection of complete Freund's adjuvant (CFA) in the hind paw. A brief (3 ms) nociceptive stimulus was successfully targeted to the contralateral (right, non-injected) paw with negligible confusion between paws (705/706 stimuli targeted the correct hind paw). This allows future studies in which phasic and tonic pain can be separated. Despite ongoing hypersensitivity and phasic stimuli, mice remained actively engaged in the task, collecting rewards sequentially from each side, as per their training (*Figure 5F and G*). Their rapid movement along the stimulation corridors (average maximum speed of 142±26 mm/s, 4 mice) required precise targeting (*Figure 5E*).

Stimulation during movement revealed a frequency-dependent disruption of path coherence and speed. High-frequency stimulation resulted in reduced locomotor speeds (*p*=0.037 with paired t-test, n=4 mice) and more variable trajectories compared to during low-frequency stimulation (*Figure 5H*). Once through the stimulation corridors, the mice successfully collected a reward in almost all trials.

We then examined how behavior depends on the behavioral state prior to stimulation. Before stimulation, two behavioral states were identified (*Figure 5I*): one represents faster locomotion in a direct heading (fast-direct state), while the other represents slower movements with headings that were less coherent (slow-assess state). The two clusters had significantly different speeds and coherence (*p*<0.0001 for both; Welch's t-test), where the 79 fast-direct trials (68.1%) showed speed of 90.8±28.3 mm/s (mean ± SD) and coherence of 0.9±0.1 (mean ± SD; arbitrary units) and the 37 slow-assess trials (31.9%) had lower speed of 32.5±13.9 mm/s and less direct headings with a coherence of 0.1±0.1. Stimulation of the two states produced robust opposing changes in behavior (*Figure 5J and K*). Slow-assess states become significantly more coherent as the stimulus drives a redirection of heading. Fast-direct states were significantly slowed and less direct, potentially representing a pause for information gain. In this case, fast-direct states shift towards slow-assess, and slow-assess shifts towards fast-assess, where intermediate states may together represent a form of vigilant behavior. This demonstrates movement state-dependence changes behavior. Indeed, the speed after stimulation was proportional to the initial speed in the fast-direct trials; a positive linear correlation (Pearson's *r*=0.23, *p*=0.046) indicates that the disruption of behavior partly scales with the initial movement state (*Figure 5L*).

We thus demonstrate automatic control of precise cutaneous stimuli as mice move through naturalistic environments.

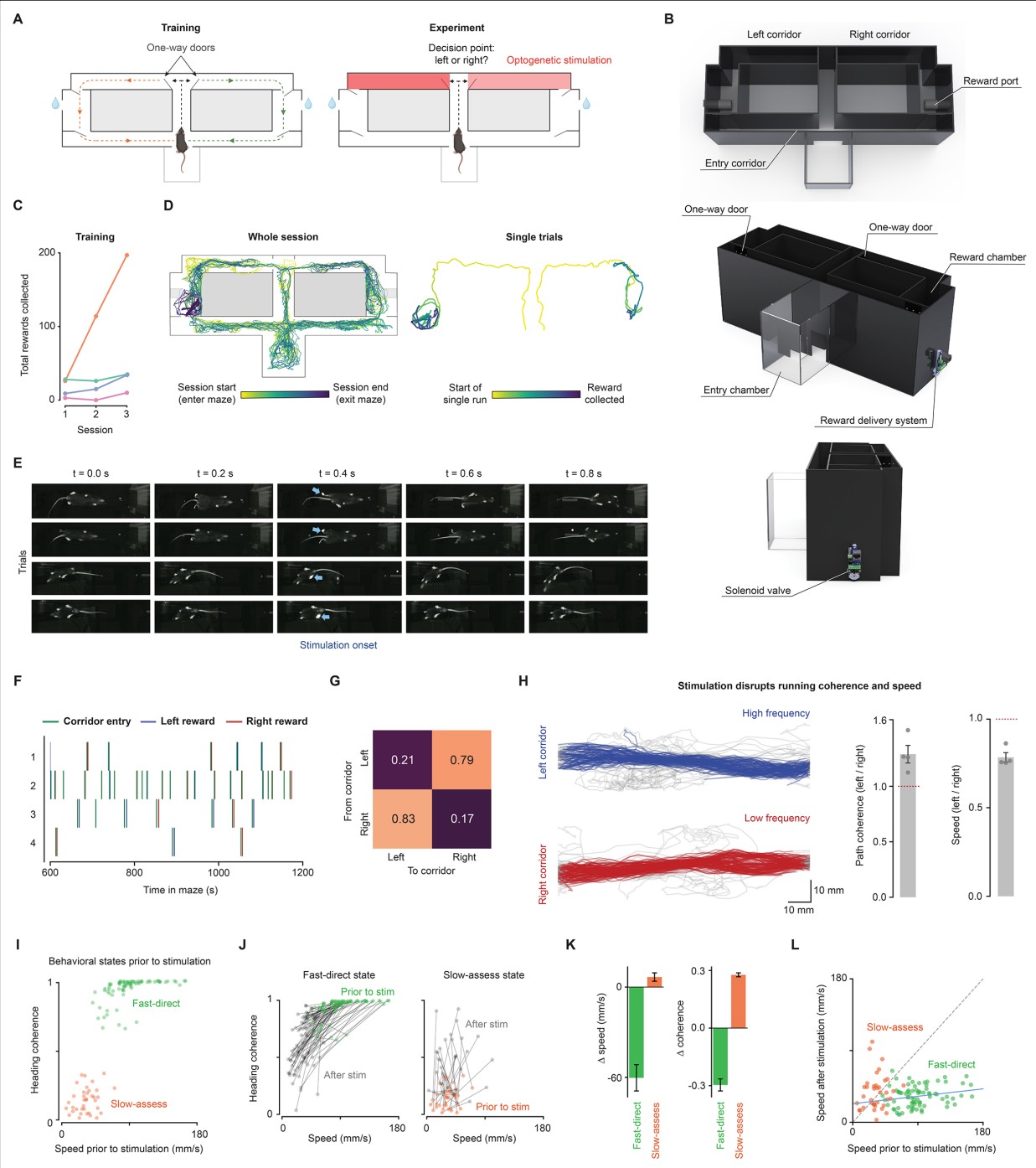

**Figure 5.** Closed-loop cutaneous stimulation in mice running through a maze. (**A**) Schematic of the maze design. A single trial was defined as the collection of a single reward, indicated by the *orange* and *green* arrows. The left and right corridors leading to the reward chambers were paired with stimulation of nociceptors using transdermal optogenetics. (**B**) Maze renderings from aerial, front, and side views. Mice entered via an entry chamber leading to a corridor and junction, choosing left or right through one-way doors. A sucrose-water reward awaited in the reward chamber, with exit through another one-way door. (**C**) Total number of rewards collected (left and right-hand side reward ports combined) for each training session. (**D**) Movement trajectories over an entire session (*left*) and a single trial (*right*). Trajectories are shown from one mouse for the first stimulation session. (**E**) Frame sequences (0.2 s apart) from four trials in four mice show runs along maze corridors toward the reward chamber. Blue arrows indicate targeted stimulation. (**F**) Relative timings of corridor entry and subsequent reward collection (n=4 mice). (**G**) Transition matrix showing mice predominately alternate between rewards at the left and right reward ports. (**H**) Example movement trajectories (tail base) in the left and right corridors from one mouse (*left*). Bar plots showing path coherence and speed in the high stimulation corridor relative to the low stimulation corridor (*right*). (**I**) Plot of speed and coherence in a 2 s window prior to the first stimulation in each trial colored by movement state cluster (Gaussian mixture modeling; 116 trials). (**J**)

*Figure 5 continued on next page*

*Figure 5 continued*

Speed and coherence relationships before and after stimulation, where the values of the prior state are colored and the responses are shown in black. The black lines pair values for each trial. Stimulation causes a consistent shift in the fast-direct state towards the slow-assess state, while stimulation of the slow-assess state shifts towards a direct movement. (**K**) Bar plots showing the change (post–pre) in speed and coherence. Stimulation resulted in changes in fast-direct trials (speed: Welch's $t=-6.90$, $p=0.006$; coherence: Welch's $t=-9.39$, $p=0.003$) and also slow-assess trials (coherence: Welch's $t=-26.9$, $p<0.0001$). (**L**) Plot of post-pre stimulation speed values across all trials colored by state clusters. Schematics in panel A were created using BioRender.com. Renderings were created using Solidworks.

## Discussion

The somatosensory system provides a critical link between the brain, the body, and its immediate external environment. The complex ways in which this system supports movement, learning, and action in rodents have historically posed substantial methodological challenges. Traditional methodologies have varied widely, encompassing both innovative and practical approaches—from stimulation of whiskers or skin in head-fixed animals (*Black et al., 2020*; *Pai et al., 2022*; *Buetfering et al., 2022*; *Guo et al., 2014*; *Hong et al., 2018*; *Campagner et al., 2019*; *Esmaeili et al., 2020*; *Schwaller et al., 2021*; *Emanuel et al., 2021*) to the more straightforward manual touching of paws in mice (*Cheng et al., 2017*; *Gangadharan et al., 2022*; *Uhelski et al., 2012*; *Gan et al., 2021*). These approaches, however, have inherent limitations in replicating the dynamic and complex interactions experienced in naturalistic environmental settings. In response to these limitations, we have developed a system that enables cutaneous stimulation in more complex environments. This closed-loop system automatically tracks, targets, and stimulates mice remotely to study the somatosensory system in naturalistic environmental settings.

Generating cutaneous inputs in freely moving mice requires stimuli that are spatially and temporally precise. We achieved millisecond-timescale stimulation of small skin areas using transdermal optogenetics ('remote touch'; *Schorscher-Petcu et al., 2021*). Opsins were genetically targeted to specific afferent fibers innervating skin and activated with light targeted precisely via a laser in free space. This beam path was aligned to a second laser system and employed for thermal stimulation on a timescale of seconds. Delivery of these stimuli was controlled by a feedback system consisting of three main components: (1) real-time tracking infers the keypoints of various body parts, continuously transmitting this data to the controller; (2) user-defined rules determine the stimulation conditions (spatial, temporal, and physiological state-dependence), providing control signals for the actuators; and (3) mirror galvanometers target the beam path to specified key points and signals trigger light delivery, following which real-time tracking is then resumed. The processing latency can be further reduced by minimizing the frame resolution, narrowing to a region of interest (ROI), increasing the processing power, using alternative networks, or combinations thereof. We demonstrate an approach to reduce the pixel count by cropping to an ROI using our random-access targeting method (*Figure 4*). One possible version of this would be centering a region on the mouse centroid as it moves. This would dramatically reduce the computational load to allow shorter processing latencies. With a reduced processing latency, high-speed video can be processed, which enables the approach to generalize to other body parts that move more rapidly. Additionally, predictive tracking can compensate for the remaining delay by estimating a keypoint location from its recent trajectory (e.g. constant-velocity extrapolation or a Kalman filter). This can reduce effective targeting error for continuously moving features, such as the snout. However, we demonstrate here that this system can precisely target the hind paw for stimulation, even as the mice are in motion.

We demonstrate the versatility of the system from automated multi-animal nociceptive testing to sparse stimulation of freely moving mice in a circular arena. We show that stimuli could be targeted with high accuracy and resulted in immediate behavioral responses that could be mapped. The system was used to deliver cutaneous stimuli to mice running through a maze. Mice were trained on an alternation task with stimuli applied en route to the reward, thus separating choice, punishment, and reward in a naturalistic environment. Mice with a model of inflammatory pain still readily engaged in this task during noxious cutaneous input (punishment). The recruitment of reflexes during locomotion caused immediate evaluative behavior that temporarily disrupted goal-directed behavior. Achieving such localized stimulation has been challenging with traditional methods: electric grid floors generate whole body stimuli that are considered incompatible with models of chronic pain that generate unilateral hind paw hypersensitivity; and manual stimulation lacks capacity, reliability, and is potentially

confounded by experimenter and observer biases. Our system addresses these issues, allowing for the dissociation of touch, phasic pain, and tonic pain to better understand their relationships with behavior.

Stimulation of the body and paws enables the study of pain, touch, thermoception, and movement (*Caterina et al., 1997*; *Decosterd and Woolf, 2000*; *Chiu et al., 2013*; *Ranade et al., 2014*; *Moy et al., 2018*; *Corder et al., 2019*; *Boyle et al., 2019*; *Juarez-Salinas et al., 2019*; *Choi et al., 2020*; *Gatto et al., 2021*; *Drake et al., 2021*; *Middleton et al., 2022*). Paw stimulation is also ubiquitous in aversive learning and memory, using a crude shock stimulation with a grid floor (*Klein et al., 2021*; *Bonapersona et al., 2022*), and can now be carried out with spatiotemporal precision. The stimulation can be static or dynamic and localized to small areas on the body in an automated manner. Automation improves the precision of stimulus delivery compared to traditional manual methods; it reduces labor and enhances the reliability of the data. All experiments were conducted remotely from an adjacent room to minimize potential observer effects and biases on the mice. Automated nociceptive assays have principally focused on the initial rapid movements elicited by stimulation of mice in small chambers (*Dedek et al., 2023*; *Iredale et al., 2023*; *Burdge et al., 2023*). Here, we provide an approach to examine how these rapid movements are embedded within complex behavior in naturalistic environments, opening new ways to investigate nociception and somatosensation more broadly.

We have previously mapped fast behavioral responses in stationary mice, demonstrating that fast responses to noxious stimuli are not localized and fixed, but are widespread and contextual (*Browne et al., 2017*) and that these are coordinated where whole-body movements depend on the initial posture (*Schorscher-Petcu et al., 2021*). Here, we show state dependence of behavioral responses in mice that are moving, where noxious stimulation causes fast-direct moving mice to slow and mice already assessing their location to shift to a direct heading. This could be considered a state-dependent modulation or switching to a form of vigilance and suggests that even the most rapid protective responses are modulated online by the current movement state, ensuring behavior remains appropriate to context.

While we demonstrate the utility of the optical system using nociceptive stimuli, this system can deliver various cutaneous inputs by targeting specific afferents for selective opsin expression, whether they are thermoreceptive, chemoreceptive, or mechanosensitive (*Schorscher-Petcu et al., 2021*; *Dedek et al., 2023*; *Iredale et al., 2023*; *Burdge et al., 2023*; *Mickle and Gereau, 2018*). Such 'pure' stimuli do not occur in nature but offer crucial spatial, temporal, and genetic precision (*Klein et al., 2021*; *Madden et al., 2016*; *Cheah et al., 2017*). Our system enables delivery of multiple wavelengths of light separately or together, to support combinations of opsins from the vast optogenetic toolbox. Opsins can be used to activate or silence neurons, with a range of kinetic properties, diverse light wavelength profiles allowing multi-color manipulations, or control different downstream signaling effectors (*Emiliani et al., 2022*). Thermal stimuli are also used routinely in research (*Pai et al., 2022*; *Dedek et al., 2023*; *Mitchell et al., 2010*) but slow thermal dissipation can require mice to be stationary for consistent stimulation. Automation provides opportunities for the development of analgesics, particularly for integrating reflexes with spontaneous, free operant behaviors. Indeed, cutaneous stimulation in naturalistic environments can be readily combined with approaches to quantify behavior (*Mathis et al., 2018*; *Jones et al., 2020*; *Wiltschko et al., 2020*; *Hsu and Yttri, 2021*; *Zhang et al., 2022*; *Weinreb et al., 2024*; *Bohic et al., 2023*; *Pereira et al., 2020*).

The system has many applications for the study of sensorimotor transformations, perception, memory, learning, and action. It is flexible enough to trigger stimulation based on various states, including periods of inactivity or locomotion, at specific spatial locations and with precise timing. Future work made possible by this system is expected to include examining how cutaneous input can interrupt and modulate specific swing phases (*Gatto et al., 2021*), self-grooming, posture states (*Browne et al., 2017*), and other spontaneous behavioral syllables (*Wiltschko et al., 2020*). It can facilitate investigations of naturalistic learning, whether through mazes, social interactions, or engagement with the environment or objects (*Choi and Kim, 2010*; *Lai et al., 2024*), and of sleep fragmentation (*Alexandre et al., 2024*), anxiety (*Orefice et al., 2019*; *La Vu et al., 2020*), fear (*Klein et al., 2021*), and stress (*Bonapersona et al., 2022*). Finally, it has the potential to provide free operant methods for analgesic development for chronic pain. The system may be combined with existing tools to record neural activity in freely moving mice, such as fiber photometry, miniscopes, or large-scale

electrophysiology, and manipulations of this neural activity, such as optogenetics and chemogenetics. This can allow mechanistic dissection of cell and circuit biology in the context of naturalistic behaviors.

Establishing how behavior is shaped by somatosensation requires that mice can be stimulated while freely behaving. We describe a system that addresses this need, delivering cutaneous stimuli in a manner that is precise, remote, state-dependent, dynamic, and fully automated to target freely behaving mice that are actively exploring complex environments.

# Methods

**Key resources table**

| Reagent type (species) or resource | Designation | Source or reference | Identifiers | Additional information |
|---|---|---|---|---|
| Strain, strain background (*Mus musculus*) | R26-CAG-LSL-hChR2(H134R)-tdTomato (Ai27D) | Jackson Laboratory | Stock #: 012567 RRID:IMSR_JAX:012567 | PMID:22446880 |
| Strain, strain background (*Mus musculus*) | Trpv1-IRES-Cre (TRPV1-Cre) | Jackson Laboratory | Stock #: 017769 RRID:IMSR_JAX:017769 | PMID:21752988 |
| Strain, strain background (*Mus musculus*) | Wild-type C57BL/6J | Jackson Laboratory | Stock #: 000664 RRID:IMSR_JAX:000664 | |
| Software, algorithm | Python | http://www.python.org/ | RRID:SCR_008394 | |
| Software, algorithm | NI-DAQmx | https://nidaqmx-python.readthedocs.io | | |
| Software, algorithm | Arduino C++ | https://www.arduino.cc | RRID:SCR_024884 | Version 1.8.18 |
| Software, algorithm | DeepLabCut | https://github.com/DeepLabCut | RRID:SCR_021391 | PMID:30127430 Version 2.2.0.2 |
| Software, algorithm | DeepLabCut-Live! | https://github.com/DeepLabCut/DeepLabCut-live | | PMID:33289631 Version 1.0 |
| Software, algorithm | Basler Pylon | https://www.baslerweb.com/en/software/pylon | | Version 6.2.4.9387 |
| Software, algorithm | Cobalt Monitor | https://hubner-photonics.com/downloads/ | | |
| Software, algorithm | Thorlabs Kinesis | https://www.thorlabs.com/kinesis-software | | |
| Software, algorithm | Original code | https://github.com/browne-lab/closed-loop-somatosensory-stimulation | RRID:SCR_028036 | See *Data availability* section |
| Software, algorithm | Seaborn | http://www.seaborn.pydata.org | RRID:SCR_018132 | |
| Software, algorithm | Adobe Illustrator | http://www.adobe.com | RRID:SCR_010279 | Version 24.0 |

## Animals

Mice were housed at 21±2°C, 55% relative humidity, following a 12 hr light: 12 hr dark cycle with *ad libitum* access to food and water. Optogenetic experiments were performed using mice with ChR2 selectively expressed in nociceptors (Trpv1::ChR2). Heterozygous Trpv1-IRES-Cre (TRPV1-Cre) mice, which have Cre recombinase inserted downstream of the *Trpv1* gene (RRID:IMSR_JAX:017769, B6.129-Trpv1$^{tm1(cre)Bbm}$/J; *Cavanaugh et al., 2011*), were crossed with mice homozygous for Cre-dependent ChR2(H134R)-tdTomato (RRID:IMSR_JAX:012567, Ai27(RCL-hChR2(H134R))/tdT-D ChR2-tdTomato; *Madisen et al., 2012*). This produced progeny heterozygous for both transgenes (Trpv1::ChR2) and control littermates that do not encode Cre recombinase but do encode Cre-dependent ChR2-tdTomato. Blue light directed to the glabrous plantar surface of the hind paw in Trpv1::ChR2 mice results in the direct time-locked activation of broad-class nociceptors with single action potential resolution (*Browne et al., 2017*). Experiments with the infrared (IR) laser were performed using wild-type mice (RRID:IMSR_JAX:000664, C57BL/6J). Equal numbers of male and female adult mice were used (aged between 6 and 40 weeks), with 2–5 cohorts of mice per experiment. Sample sizes reflected common practice in the field, and exact *n* is provided in figure legends. Conditions were randomized and experiments were performed blind to group allocation. Predefined exclusion criteria were applied

only when stated (e.g. insufficient maze-task engagement). All animal work was carried out according to the UK Animal Scientific Procedures Act (1986), approved by the UCL Animal Welfare and Ethical Review Body (AWERB) and performed under licenses released by the UK Home Office.

## Design and development

Several substantial improvements were made to the optical design (*Schorscher-Petcu et al., 2021*) to enable automated, multi-color, closed-loop optical stimulation across a large environment. Part lists are provided in *Supplementary files 1–3*.

The optical system was mounted on a large aluminum breadboard (0.75 m×0.75 m) to provide more space for optical components and stability to the large glass platform. The diode laser beam (blue light, 473 nm, Cobolt, 06–01 MLD) was focused to the center of the galvanometers using two broadband dielectric mirrors (M1 and M2) via an axial adjustable lens (L1, 30 mm focal length), a collimating lens (L2, 150 mm focal length), and a long focal length lens (L3, 500 mm focal length). We added a second laser beam path to enable multi-color stimulation, using separate mirrors and lenses and an appropriate dichroic mirror. The infrared (IR) laser (785 nm, SLOC, RLM785TA-1500) beam passed through an optical beam shutter (Thorlabs, SH05RM) to pulse the light with a controller (Thorlabs, KSC101). Two additional mirrors (M3 and M4) aligned the IR beam through a long focal length lens (750 mm) to the DM, where the beam path was aligned to converge with the blue light laser beam path into a pair of galvanometer mirrors (GM).

For the large environment, a 0.55 m×0.55 m glass stimulation platform was held in place above the optical components via a vertical optical construction rail (95 mm×95 mm×1500 mm) attached to the aluminum breadboard, as shown in *Figure 2A*. Aluminum construction rails (25 mm×25 mm× 500 mm) were secured at each corner of the glass platform frame and the opposite side of the platform to the optical rail to ensure stability. The blue light laser spot size (1/e$^2$ width) was calibrated to 2.3 mm$^2$ using the non-rotating L1 adjustable lens housing and an optical beam profiler (BP209-VIS/M, Thorlabs). For the experiment using the IR laser, two near-IR hot mirrors (Thorlabs, FM201) were placed on top of the USB 3.0 camera (acA1920-40um camera, Basler) lens to minimize how much IR light was imaged by the camera.

For real-time markerless pose estimation to support automated, closed-loop stimulation, an additional camera was positioned below the glass stimulation platform. Behavior was captured at 30 frames per second (fps) via a USB 3.0 to the primary computer (C1), which controls video recording, pose estimation, calculations, and directs the galvanometer mirrors to target lasers.

## Optical system calibration

The optical parameters of the system were characterized using the blue laser due to the high-quality beam. The uniformity of blue light diode laser spot size across the glass stimulation platform was measured with an optical beam profiler (Thorlabs, BP209-VIS/M) placed at 16 locations across the platform. The beam profiler aperture was positioned at these locations using a custom laser-cut acrylic plate. Laser power was attenuated by 25% with an ND filter (Thorlabs, NE506B, optical density 0.6) to be within the operating range of the beam profiler. Absolute power (mW) at the 16 locations was assessed with a S121C photodiode measured by an optical power meter (Thorlabs, PM100D). The laser beam area and the optical power (mW) were calculated at each location (*Figure 2—figure supplement 1A*).

There was negligible distortion in the acquisition camera across the glass platform. This was determined by imaging a chessboard camera calibration pattern of 20 mm×20 mm squares in a 14×10 grid at five different locations across the glass. OpenCV was used to measure square sizes, and we calculated the min-max range of all squares was <1 pixel, at 0.89 pixels, which is considered negligible. The Euclidean norm was computed for a matrix of the corners of all squares, providing a scale factor of 0.45 mm/pixel.

To generate a pixel-voltage coordinate dictionary that can be used to convert $x, y$ pixel coordinates to $x, y$ galvanometer voltage coordinates, the following steps were carried out. First, the galvanometers were raster stepped to direct the blue laser spot to a grid of 10,000 points (100×100), capturing these with the pose-estimation camera. For every point of the raster, the $x, y$ voltages were mapped to the peak intensity pixel. The $x, y$ voltages were then computed for every pixel by interpolation, fitting with a two-dimensional polynomial equation. This automated procedure took <30 min and resulted

in a pre-computed pixel-voltage dictionary. Entering an *x, y* coordinate for a body part, inferred from the camera feed, returns the interpolated *x, y* voltages to target the laser to the same location. We repeated the mapping once every week over the course of 10 weeks to ensure the stability of the mapping. This was done during extensive experimentation to account for potential movements during cleaning and changes in arenas.

## Pose estimation

### Training a DeepLabCut network model

DeepLabCut installation (v2.2.0.2; *Mathis et al., 2018*; *Nath et al., 2019*) was coupled to Tensor-Flow-GPU (v2.5.0, with CUDA v11.2 and cuDNN v8.1). Training of the DeepLabCut neural network model was used with default network and training settings in an Anaconda environment with Python v3.8.13 installed. Videos were selected based on their representation of the whole breadth of behavioral responses, and k-means clustering was used to select the training images. 437 frames were labeled from 22 selected videos, and the network was trained for 200,000 iterations. Following further optimization of lighting, 210 frames from 11 additional videos were manually labeled, and machine labels from 171 outlier frames from nine videos were manually refined. These were fed back to the training dataset and the network retrained for a further 200,000 iterations. Training resulted in an MAE of 3.29 pixels, which is comparable to human ground truth variability quantified elsewhere (see *Mathis et al., 2018*). This model was used for all pose estimation. The video resolution (1920×1200) required a processing time longer than the frame interval (33.33 ms), resulting in real-time pose estimation on a sub-sample of all frames recorded. Therefore, *post-hoc* pose estimation was carried out to analyze all frames.

### Real-time tracking

DLC-Live! SDK (v1.0; *Kane et al., 2020*) was installed on a computer with fast processing capabilities (AMD Ryzen 5 3600 six-core CPU (3.6–4.2 GHz), NVIDIA GeForce RTX 2080 Ti GPU, 64 GB RAM, Windows 10, custom manufactured by PC Specialist Ltd.) in an Anaconda virtual environment (Python v3.7.10) with DeepLabCut (v2.1.10.4) installed. DLC-Live! SDK installation was coupled to Tensor-Flow-GPU (v1.13, with CUDA v10 and cuDNN v7.4). Integration of the Basler camera and the DLC-Live! GUI (DLG) utilized a Python wrapper, pypylon (v1.7.2, Basler), to facilitate communication with the pylon Camera Software Suite through a Linux subsystem in Windows 10 (WSL Ubuntu, v20.04). The trained DeepLabCut network model was loaded into the DLG, which captures the data from the camera and performs real-time pose estimation on the incoming camera feed. Custom code was written in Python for each experimental design; this comprised the conditions that defined the behavioral protocol and controlled stimulation as required (see *Figure 2*).

## Optical system characterization

We characterized the latencies for real-time tracking, targeting, and stimulation. Control signals for the camera, mirror galvanometers, and laser were measured simultaneously at 100 kHz using a Digidata 1440 a (Molecular Devices). During the exposure of each 5 ms frame, the tracking camera sent a voltage signal from its GPIO. The *x-* and *y*-axis scanner position outputs from the two mirror galvanometer drivers were used to monitor the movement of the mirrors. A 1 ms laser signal was sent to a microcontroller (Arduino Uno) to generate parallel digital outputs, which triggered the laser and monitored its timings. All four control signals were recorded during four 5 min sessions with wild-type mice (C57BL/6J) exploring a circular arena. The tracking camera was set to record at a resolution of 1100 pixels×1100 pixels, with a 5 ms exposure at 30 fps. The processor code identified frames with a likelihood >0.8 for the 'left_hindpaw_mid' keypoint. The *x, y* pixel location was then converted to mirror galvanometer *x, y* voltage signals using a pixel-voltage coordinate dictionary. A multifunction DAQ device (USB-6002, National Instruments) was used to send these *x, y* voltages and subsequently send a 1 ms command to the laser-triggering microcontroller. The laser was triggered only if more than 500 ms had passed since the previous stimulation. The camera signal confirmed an exposure time of 5 ms and a frame rate of 30 fps. The latency between camera acquisition and stimulation was calculated by collecting timestamps immediately after stimulation (acquisition timestamp) and comparing these to the frame timestamp on which pose estimation was carried out (processing timestamp). We estimated a processing latency of 84±12 ms (mean ± SD)

by subtracting the frame acquisition timestamp from the frame processing timestamp contained in DeepLabCut output files for 16,000 processed frames recorded across four mice (4000 frames each). The latency between galvanometers moving and laser stimulation was determined by comparing the timings of galvanometer jumps and laser signals. This delay was 3.3±0.5 ms (mean ± SD, for 245 trials). To synchronize the four voltage signals with frame and stimulation timestamps, we determined the timing of the first galvanometer jump when pose estimation was initialized. In the current configuration, the end-to-end closed-loop delay is ≈87 ms from the combination of the processing latency and other delays.

The accuracy of real-time tracking for the 'left_hindpaw_mid' keypoint was assessed by manually identifying its coordinates (ground truth) and comparing these to the coordinates predicted by the DeepLabCut network model in real time on frames extracted from five videos of different mice exploring an open arena. Frames with a likelihood >0.8 were selected, as in experimental protocols. Euclidean distances were calculated pairwise between ground truth coordinates and model-generated coordinates and averaged to give the mean average Euclidean error (MAE). The MAE between the predicted and actual coordinates was 1.36 mm (calculated on 1281 frames).

The accuracy of body part targeting was determined using a high-speed Basler acA2000-165μm NIR camera recording frames at 648×650 pixels, 270 fps during the 5 min sessions described above. We used a>0.8 likelihood for the 'left_forepaw' keypoint in additional sessions. High-speed recordings captured each 1 ms laser pulse, and frames containing these pulses were identified using the reflection of the laser. We manually assessed 1279 frames and classified them as a 'hit' or 'miss,' and whether a 'hit' was on the targeted paw to quantify confusion during keypoint tracking.

The accuracy of hitting the body part depended on how fast the mice were moving. To demonstrate this, we segmented the keypoint series using four speed categories: stationary, low, medium, and high. Speed was calculated using the Euclidean distance the 'tail_base' keypoint moved in each frame, dividing this by the time elapsed and smoothing the speed with a 10-frame rolling mean filter. The speed histogram informed the windows of categories (<20, 20–120, 120–220, >220 pixels per second, for stationary, low, medium, and high, respectively). The accuracy of hitting the 'left_hindpaw_mid' keypoint was calculated on frames across each speed category: 156 frames for stationary, 159 frames for low, 155 frames for medium, and 187 frames for high. Similarly, the accuracy of hitting the 'left_forepaw' keypoint was calculated on 155 frames for stationary, 156 frames for low, 155 frames for medium and 156 frames for high.

The error between the ground truth keypoint and the laser spot for the 'left_hindpaw_mid' keypoint was determined by manually identifying the body part coordinates (ground truth) on frames immediately prior to stimulation and then the coordinates for the laser spot on subsequent stimulation frames. These estimates were first made for all pre-stimulation frames and then for the set of stimulation frames. The mean average Euclidean error (MAE) was approximately 1.3 mm across all locomotion speed categories (463 frames).

## Multi-chamber real-time pose estimation

To target and stimulate individual mice when multiple mice were present in chambers on the stimulation platform, we performed chamber-based cropping and subsequent real-time pose estimation. Nine mice were placed into nine chambers (100 mm×100 mm wide, 120 mm tall). We monitored the motion in each chamber to find mice that were 'idle'. The camera feed was cropped, body parts estimated, and the laser targeted to the hind paw coordinates.

The frame-to-frame absolute difference in pixel values (motion energy) was calculated in each region of interest for the individual chambers. Background noise was removed below <10 motion energy, and the mouse was defined as 'idle' if the summed motion energy was less than a specified threshold (30,000 motion energy) for 2 s. Idle mice that had not been stimulated in the previous 10 s were pseudo-randomly selected and their chamber cropped. The pose estimation (*x*, *y*) coordinates generated by the DeepLabCut network model were used to target the laser to the hind paw. We modified the following scripts in the *dlclive* and *dlclivegui* packages in DLC-Live! SDK to develop the multi-chamber real-time tracking approach: *dlclive*, *utils* (*dlclive* package) and *pose_process* (*dlclivegui* package).

## Assembly of a naturalistic task

The maze was constructed of 3 mm matte black acrylic (200 mm in height) and measured 500 mm×180 mm (inner dimensions). The maze was constructed as a single junction, with 40 mm width corridors forming two chambers (70 mm×100 mm) at either end of the junction corridors. The entrance to the maze was connected to a transparent acrylic chamber (100 mm×100 mm, 130 mm tall). There were one-way doors (100 mm tall, 30 mm wide) designed as a push-through flap cut from 0.5 mm styrene and secured to the door frame with butterfly pins. The one-way doors were positioned at the junction and to leave the reward chambers; this created a one-way system, so once the mouse exited either chamber, it was required to go back around the maze and through the junction decision point to re-enter the reward chamber. Each chamber contained a rectangular opening (20.5 mm×11.5 mm) through which a water delivery port (Sanworks mouse behavior port) was fixed to the walls to allow the mouse to collect rewards. A water reward (~5 µl of 10% sucrose water) was delivered when the mouse's nose broke the IR beam in the water delivery port. The reward delivery system was controlled with an Arduino. In addition to a reward timeout period of 45–60 s, the mouse was required to leave the chamber before the water reward port was reset and another reward could be collected.

## Behavioral protocols

### Experimental room, arena, and cleaning set-up

The experimental room was maintained at 21°C with relative humidity between 45–65%. All behavior experiments on the system were performed in custom-built arenas laser cut from matte black acrylic and placed on the glass stimulation platform. Two infrared LED panels illuminated opposite sides of the arena to optimize lighting and achieve high-contrast images. White noise at 68 dB was generated with custom Python code, through a L60 Ultrasound Speaker (Petterson Elektronic AB) via a second DAQ device (USB-6211, NI) and amplifier. The white noise played continuously through the duration of the habituation sessions and the experiment. The glass stimulation platform was cleaned twice with 70% ethanol, while the acrylic arena was cleaned twice with an odorless surface disinfectant between each animal to minimize olfactory cues. The lasers were targeted to the hind or fore paw glabrous skin in all experiments, contingent on meeting specific conditions defined in the protocol.

### Habituation

Animals were placed in custom matte black acrylic chambers (100 mm×100 mm, 80 mm in height) placed on a von Frey wire mesh grid and underwent two habituation sessions to the experimental room for 1–2 hr. Mice also underwent 1 or 2 handling sessions prior to experiments.

### Minimal cutaneous stimulation in an open arena

Mice were placed in an acrylic arena painted matte black (500 mm outer diameter, 150 mm in height, 5 mm thick). Dividers (160 mm tall, 116 mm wide, matte black, 3 mm thick) were slotted onto the arena wall to separate the arena into six segments to enrich the environment. Mice were allowed to freely explore for 60 min. Individual 10 ms duration blue light laser pulses were remotely targeted to the left hind paw with a≥10 min inter-stimulus interval. Each stimulation was delivered contingent on the conditions that the hind paw was still and had not been stimulated within the previous 10 min.

The hind paw was considered still when both the standard deviation of its keypoint (*x, y*) was <1 pixel and the likelihood of this keypoint was >0.8 throughout a 2 s period. Stimulation was repeated over two sessions on consecutive days. Data were collected from 26 mice in total from five different cohorts. 16 Trpv1::ChR2 mice were split into two groups: 10 mice received blue light stimulation, and 6 mice received no stimulation as a control. Ten littermate controls that received blue light stimulation were also used.

### Somatosensory stimulation in multiple chambers

Nine mice were placed in 100 mm x 100 mm individual chambers in a 3×3 configuration, covered by a lid. Mice were habituated to the chambers atop the glass stimulation platform for 2 hr in two sessions prior to the first experimental day.

For the experiment with thermal stimulation, 18 C57BL/6J mice from 2 cohorts were used. Mice were placed in the chambers for 2 hr, and a 10 s laser pulse was targeted to one of the hind paws,

with up to 10 stimulations on each paw >1 min apart. IR laser spot size was 2.2 mm$^2$ and the optical power was set to 1.4 W in the first cohort of mice and to 1.65 W in the second cohort of mice to elicit paw responses between 10–12 s.

For the experiment with transdermal optogenetic stimulation, 9 Trpv1::ChR2 and 9 littermate controls from two cohorts were used. Mice were similarly placed in the chambers for 2 hr, with optogenetic stimulations delivered to each hind paw >1 min apart. The stimulation protocol comprised six conditions: a single pulse stimulation at 40 mW/mm$^2$, and a train of pulses (3 ms pulses at 10 Hz for 10 s) at 8, 4, 2, 1, and 0.5 mW/mm$^2$. Spot size was 2.0 mm$^2$. The order of stimulation intensity was pseudorandomized with Euler tours (**Bakermans and Behrens, 2021**).

### Somatosensory stimulation in a maze

Mice were first habituated to the maze without any doors during a 1 hr session. On three separate days following this, they underwent 3 training sessions with the one-way doors in place. Mice were water-deprived 16–18 hr prior to each experimental session to motivate the use of water rewards. A trial was defined as the mouse successfully collecting one reward; the collection of multiple rewards required the mouse to leave the reward chamber. Mice that had not made >10 trials by the third training session were excluded from the subsequent stimulation sessions due to poor engagement. Seven out of 12 Trpv1::ChR2 mice from four cohorts met this criterion. As proof of principle for precise contralateral stimulation in the context of a unilateral pain state, mice received 7 $\mu$l of complete Freund's adjuvant (CFA) via intraplantar injection in the left hind paw. CFA-injected mice showed significant mechanical allodynia compared to saline controls ($p=0.039$ with the Mann-Whitney test, n=4 mice). After baseline measurements of mechanical sensitivity, mice were injected with CFA, and mechanical allodynia was evaluated in both hind paws 2 days following injection. Mechanical allodynia resulting from injection of CFA into the left hind paw was measured by von Frey testing (Up-Down method). Mice were placed in individual chambers (100 mm×100 mm) on a mesh wire floor and habituated to the test setup prior to testing. The von Frey test was conducted blind to experimental groups. Mice underwent two stimulation sessions in the maze, in which optogenetic stimuli were delivered to the right (uninjected) hind paw in the stimulation zones. There were two stimulation protocols: the left corridor was paired with 3 ms laser pulses at 5 Hz and the right corridor was paired with 3 ms laser pulses at 1 Hz. Laser power density was 40 mW/mm$^2$. Training and experimental sessions lasted 1–2 hr.

## Data analysis

### Data compression, analysis, and visualization

Videos were acquired in AVI format and fed through offline DeepLabCut pose estimation to generate ($x$, $y$) coordinates and likelihoods for each body part. For the analysis of the recordings with multiple chambers, AVI video files were converted to MP4 format using H.264 compression. The MP4 video files were cropped into individual mouse chambers (230×230 pixels) before running pose estimation. Analyses were based on the position of the hind paw or tail base coordinates. All analysis code was written in Python 3 (v3.9.7), using the NumPy, Pandas, and OpenCV packages. Data was visualized using Matplotlib and Seaborn packages.

### Calculation of paw response latency with motion energy

Local motion energy was calculated inside a circular ROI (radius = 15 px) centered on the stimulation-side paw keypoint by taking absolute differences between consecutive frames, masking the ROI, suppressing values <3 intensity units, and summing the remaining pixels. The ROI center was anchored to the DLC keypoint at the frame immediately before the nominal stimulation onset. The nominal stimulus time was refined per trial by locating the largest motion-energy peak within ±0.2 s of nominal onset; traces were then time-locked to this real onset and re-windowed from –9.8 to +9.8 s, with the single-frame flash artifact at $t$=0 set to zero. Motion-energy traces used for detection were denoised with a 3-sample median followed by a 7-sample Gaussian ($\sigma$=2) before thresholding. Trials with threshold crossings in the 0–1.5 s window were not considered sufficiently idle. A trial was considered a response if the stimulation-paw motion energy first exceeded a fixed threshold ($\geq$1000 summed units) within the analysis window of 1.5–9.8 s. Response latency was defined, for responding trials, as the time of the first threshold crossing within this window minus the refined onset time.

## Calculation of paw response latency with pose estimation

We calculated the Euclidean distances for the paw keypoint during the trial window (–2 to +10 s; optogenetic stimulus at $t$=0) relative to its mean baseline coordinate (–2–0 s). Analysis was conducted on the keypoint on the hind paw toes to reduce stimulation artifacts from light delivered to the center of the hind paw using coordinates with >0.8 likelihood values. If the keypoint moved more than three pixels within the stimulation trial window, the trial was classified as a response. For the responses, the latency was determined by taking the time where the movement first exceeds three pixels, relative to the stimulation onset time.

## Calculation of speed

The estimated tail base coordinates were used to visualize trajectories in the open arena and maze. These estimated coordinates were used if the likelihood >0.8 (open arena) and >0.85 (maze). Tracking errors were removed when the Euclidean distance jumped >30 pixels in a single frame, and linear interpolation was performed using the three frames either side of the removed values. Speed was calculated by taking the difference in Euclidean distance ($\Delta d$) between frames as a function of the respective difference in frame times ($\Delta t$) and converted to mm/s using the scale factor calculated above. To reduce frame-to-frame jitter and suppress tracking noise while preserving changes in movement, the speed trace was smoothed using a rolling median filter with a window of 10 samples. For the maze, we calculated the speed (vigor) by capturing each 'corridor run' from the point the tail base entered the corridor to when it exited the corridor. Speed for the corridor run was calculated within this time window as above.

## Calculation of movement state dependence

Movement state clusters (fast-direct and slow-assess) were identified from tail-base speed and heading coherence in the 2 s window prior to stimulation. Speed was determined as Euclidean displacement divided by the frame interval, smoothed with a 10-frame rolling median, and the mean over the 2 s window was taken. Heading coherence was calculated from the frame-to-frame heading angle of the tail-base keypoint: headings were circularly smoothed by averaging unit vectors over a 10-frame window, and coherence was defined as the mean resultant length ($R$, 0–1) over the 2 s window. Plotting speed and coherence for the first stimulus hit in each trial revealed two discrete populations, consistent across all four mice. Gaussian mixture modeling assigned trials to two clusters whose speed and coherence differed significantly. Taking speed and coherence in the 2 s post-stimulation window allowed paired analysis within trials to evaluate how the behavioral change depends on its prior state.

## Statistical analysis

Statistical analysis was performed in Python, with the SciPy, Statsmodels, and Pingouin packages. Normality was determined using the Shapiro-Wilk normality test. The specific tests used for each comparison are detailed in the text. Statistical significance was considered as $p<0.05$. Data are reported as mean ± standard error of the mean (SEM) unless stated otherwise. The mouse was the experimental unit.

## Materials availability

System configurations will be made available upon reasonable request to L.E.B.

## Acknowledgements

We are grateful to Patrick Haggard and Andrew MacAskill for their comments on the initial manuscript. This work was supported by a Sir Henry Dale Fellowship jointly funded by the Wellcome Trust and the Royal Society (109372/Z/15/Z) and funding from the Medical Research Council (MR/N013867/1).

# Additional information

## Funding

| Funder | Grant reference number | Author |
|---|---|---|
| Wellcome Trust and the Royal Society | 10.35802/109372 | Liam E Browne |
| Medical Research Council | MR/N013867/1 | Isobel Parkes |

The funders had no role in study design, data collection and interpretation, or the decision to submit the work for publication. For the purpose of Open Access, the authors have applied a CC BY public copyright license to any Author Accepted Manuscript version arising from this submission.

## Author contributions

Isobel Parkes, Data curation, Software, Formal analysis, Validation, Investigation, Visualization, Methodology, Writing – original draft, Writing – review and editing; Ara Schorscher-Petcu, Qinyi Gan, Methodology, Writing – review and editing; Liam E Browne, Conceptualization, Resources, Software, Formal analysis, Supervision, Funding acquisition, Validation, Investigation, Visualization, Methodology, Writing – original draft, Project administration, Writing – review and editing

## Author ORCIDs

Isobel Parkes ⓘ https://orcid.org/0000-0002-6569-618X
Ara Schorscher-Petcu ⓘ https://orcid.org/0000-0001-5808-5172
Qinyi Gan ⓘ https://orcid.org/0009-0006-3962-5501
Liam E Browne ⓘ https://orcid.org/0000-0002-5693-7703

## Ethics

All animal work was carried out according to the UK Animal Scientific Procedures Act (1986), approved by the UCL Animal Welfare and Ethical Review Body (AWERB) and performed under licenses released by the UK Home Office.

Reviewer #1 (Public review): https://doi.org/10.7554/eLife.106033.3.sa1
Reviewer #2 (Public review): https://doi.org/10.7554/eLife.106033.3.sa2
Reviewer #3 (Public review): https://doi.org/10.7554/eLife.106033.3.sa3
Author response https://doi.org/10.7554/eLife.106033.3.sa4

# Additional files

## Supplementary files

Supplementary file 1. Optics. This table details the optical components for the assembly of the system.

Supplementary file 2. Mounting components. This table details the parts for mounting optics in the system.

Supplementary file 3. Acquisition and control components. This table details the parts for acquisition and control.

MDAR checklist

## Data availability

The manuscript is a Tools and Resources study that provides a new method. Code to use this method is openly accessible at https://github.com/browne-lab/closed-loop-somatosensory-stimulation (*Parkes et al., 2026*). Data is available on Dryad at https://doi.org/10.5061/dryad.rr4xgxdhs.

The following dataset was generated:

| Author(s) | Year | Dataset title | Dataset URL | Database and Identifier |
|---|---|---|---|---|
| Parkes I, Schorscher-Petcu A, Gan Q, Browne LE | 2026 | Precision cutaneous stimulation in freely moving mice | https://doi.org/10.5061/dryad.rr4xgxdhs | Dryad Digital Repository, 10.5061/dryad.rr4xgxdhs |

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
