## [Editor Report · eLife Assessment]

This **important** study combines real-time key point tracking with transdermal activation of sensory neurons as a general technique to explore how somatosensory stimulation impacts behavior in freely moving mice. After addressing concerns about classification of the behavioral responses to nociceptor stimulation, the authors now **convincingly** demonstrate a state-dependence in the behavioral response following nociceptor activation, highlighting how their real-time optogenetic stimulation capabilities can yield new insights into complex sensory processing. This work is a technological advancement that will be of interest to a broad readership, in particular labs studying somatosensation, enabling rigorous investigation of behaviors that were previously difficult or impossible to study.

---

## [Referee Report · Reviewer #1 (Public review)]

Summary:

This study presents a system for delivering precisely controlled cutaneous stimuli to freely moving mice by coupling markerless real-time tracking to transdermal optogenetic stimulation, using the tracking signal to direct a laser via galvanometer mirrors. The principal claims are that the system achieves sub-mm targeting accuracy with a latency of <100 ms. Due to the nature of mouse gait, this enables accurate targeting of forepaws even when mice are moving.

Strengths:

The study is of high quality and the evidence for the claims is convincing. There is increasing focus in neurobiology in studying neural function in freely moving animals, engaged in natural behaviour. However, a substantial challenge is how to deliver controlled stimuli to sense organs under such conditions. The system presented here constitutes notable progress towards such experiments in the somatosensory system and is, in my view, a highly significant development that will be of interest to a broad readership.

My comments on the original submission have been fully addressed.

---

## [Referee Report · Reviewer #2 (Public review)]

Parkes et al. combined real-time keypoint tracking with transdermal activation of sensory neurons to examine the effects of recruitment of sensory neurons in freely moving mice. This builds on the authors' previous investigations involving transdermal stimulation of sensory neurons in stationary mice. They illustrate multiple scenarios in which their engineering improvements enable more sophisticated behavioral assessments, including (1) stimulation of animals in multiple states in large arenas, (2) multi-animal nociceptive behavior screening through thermal and optogenetic activation, and (3) stimulation of animals running through maze corridors. Overall, the experiments and the methodology, in particular, is written clearly. The revised manuscript nicely demonstrates a state-dependence in the behavioral response to activation of TrpV1 sensory neurons, which is a nice demonstration of how their real-time optogenetic stimulation capabilities can yield new insights into complex sensory processing.

Comments on revisions:

I agree that your revisions have substantially improved the clarity and quality of the work.

---

## [Referee Report · Reviewer #3 (Public review)]

Summary:

To explore the diverse nature of somatosensation, Parkes et al. established and characterized a system for precise cutaneous stimulation of mice as they walk and run in naturalistic settings. This paper provides a framework for real-time body part tracking and targeted optical stimuli with high precision, ensuring reliable and consistent cutaneous stimulation. It can be adapted in somatosensation labs as a general technique to explore somatosensory stimulation and its impact on behavior, enabling rigorous investigation of behaviors that were previously difficult or impossible to study.

Strengths:

The authors characterized the closed-loop system to ensure that it is optically precise and can precisely target moving mice. The integration of accurate and consistent optogenetic stimulation of the cutaneous afferents allows systematic investigation of somatosensory subtypes during a variety of naturalistic behaviors. Although this study focused on nociceptors innervating the skin (Trpv1::ChR2 animals), this setup can be extended to other cutaneous sensory neuron subtypes, such as low-threshold mechanoreceptors and pruriceptors. This system can also be adapted for studying more complex behaviors, such as the maze assay and goal-directed movements.

Weaknesses:

Although the paper has strengths, its weakness is that some behavioral outputs could be analyzed in more detail to reveal different types of responses to painful cutaneous stimuli. For example, paw withdrawals were detected after optogenetically stimulating the paw (Figures 3E and 3F). Animals exhibit different types of responses to painful stimuli on the hindpaw in standard pain assays, such as paw lifting, biting, and flicking, each indicating a different level of pain. The output of this system is body part keypoints, which are the standard input to many existing tools. Analyzing these detailed keypoints would greatly strengthen this system by providing deeper biological insights into the role of somatosensation in naturalistic behaviors. Additionally, if the laser spot size could be reduced to a diameter of 2 mm², it would allow the activation of a smaller number of cutaneous afferents, or even a single one, across different skin types in the paw, such as glabrous or hairy skin.

Comments on revisions:

The authors successfully addressed all of my questions and concerns.

---

## [Author Response]

The following is the authors’ response to the original reviews.

**Public Reviews:**
**Reviewer #1 (Public review)**:Summary:This study presents a system for delivering precisely controlled cutaneous stimuli to freely moving mice by coupling markerless real-time tracking to transdermal optogenetic stimulation, using the tracking signal to direct a laser via galvanometer mirrors. The principal claims are that the system achieves sub-mm targeting accuracy with a latency of <100 ms. The nature of mouse gait enables accurate targeting of forepaws even when mice are moving.Strengths:The study is of high quality and the evidence for the claims is convincing. There is increasing focus in neurobiology in studying neural function in freely moving animals, engaged in natural behaviour. However, a substantial challenge is how to deliver controlled stimuli to sense organs under such conditions. The system presented here constitutes notable progress towards such experiments in the somatosensory system and is, in my view, a highly significant development that will be of interest to a broad readership.Weaknesses:(1) "laser spot size was set to 2.00 } 0.08 mm2 diameter (coefficient of variation = 3.85)" is unclear. Is the 0.08 SD or SEM? (not stated). Also, is this systematic variation across the arena (or something else)? Readers will want to know how much the spot size varies across the arena - ie SD. CV=4 implies that SD~7 mm. ie non-trivial variation in spot size, implying substantial differences in power delivery (and hence stimulus intensity) when the mouse is in different locations. If I misunderstood, perhaps this helps the authors to clarify. Similarly, it would be informative to have mean & SD (or mean & CV) for power and power density. In future refinements of the system, would it be possible/useful to vary laser power according to arena location?

We thank the reviewer for their comments and for identifying areas needing more clarity. The previous version was ambiguous: 0.08 refers to the standard deviation (SD). We have removed the ambiguity by stating mean ± SD and reporting a unitless coefficient of variation (CV).

The revised text reads “laser spot size was set to 2.00 ± 0.08 mm^2^ (mean ± SD; coefficient of variation = 0.039).” This makes clear that the variability in spot size is minimal: it is 0.08 mm^2^ SD (≈0.03 mm SD in diameter). This should help clarify that spot size variability across the arena is minute and unlikely to contribute meaningfully to differences in stimulus intensity across locations. The power was modulated depending on the experiment, so we provide the unitless CV here in “The absolute optical power and power density were uniform across the glass platform (coefficient of variation 0.035 and 0.029, respectively; Figure 2—figure supplement)”. We are grateful to the reviewer for spotting these omissions.

The reviewer also asks whether, in the future, it is “possible/useful to vary laser power according to arena location”. This is already possible in our system for infrared cutaneous stimulation using analog modulation (Figure 4). We have added the following sentence to make this clearer: “Laser power could be modulated using the analog control.”

(2) "The video resolution (1920 x 1200) required a processing time higher than the frame interval (33.33 ms), resulting in real-time pose estimation on a sub-sample of all frames recorded". Given this, how was it possible to achieve 84 ms latency? An important issue for closed-loop research will relate to such delays. Therefore please explain in more depth and (in Discussion) comment on how the latency of the current system might be improved/generalised. For example, although the current system works well for paws it would seem to be less suited to body parts such as the snout that do not naturally have a stationary period during the gait cycle.

We captured and stored video with a frame-to-frame interval of 33.33 ms (30 fps). DeepLabCut-live! was run in a latency-optimization mode, meaning that new frames are not processed while the network is busy - only the most recent frame is processed when free. The processing latency is measured per processed frame, and intermediate frames are thus skipped while the network is busy. Although a wide field of view and high resolution is required to capture the large environment, increasing the per-frame compute time, the processing latency remained small enough to track and stimulate moving mice. This processing latency of 84 ± 12 ms (mean ± SD) was calculated using the timestamps stored in the output files from DeepLabCut-live!: subtracting the frame acquisition timestamp from the frame processing timestamp across 16,000 processed frames recorded across four mice (4,000 each). In addition, there is a small delay to move the galvanometers and trigger the laser, calculated as 3.3 ± 0.5 ms (mean ± SD; 245 trials). This is described in the manuscript, but can be combined with the processing latency to indicate a total closed-loop delay of ≈87 ms so we have expanded on the ‘Optical system characterization’ subsection in the Methods, adding “We estimated a processing latency of 84 ± 12 ms (mean ± SD) by subtracting…” and that “In the current configuration the end-to-end closed-loop delay is ≈87 ms from the combination of the processing latency and other delays”. To the Discussion, we now comment on how this latency can be reduced and how this can allow for generalization to more rapidly moving body parts.

**Reviewer #2 (Public review)**:Parkes et al. combined real-time keypoint tracking with transdermal activation of sensory neurons to examine the effects of recruitment of sensory neurons in freely moving mice. This builds on the authors' previous investigations involving transdermal stimulation of sensory neurons in stationary mice. They illustrate multiple scenarios in which their engineering improvements enable more sophisticated behavioral assessments, including (1) stimulation of animals in multiple states in large arenas, (2) multi-animal nociceptive behavior screening through thermal and optogenetic activation, and (3) stimulation of animals running through maze corridors. Overall, the experiments and the methodology, in particular, are written clearly. However, there are multiple concerns and opportunities to fully describe their newfound capabilities that, if addressed, would make it more likely for the community to adopt this methodology:The characterization of laser spot size and power density is reported as a coefficient of variation, in which a value of ~3 is interpreted as uniform. My interpretation would differ - data spread so that the standard deviation is three times larger than the mean indicates there is substantial variability in the data. The 2D polynomial fit is shown in Figure 2 - Figure Supplement 1A and, if the fit is good, this does support the uniformity claim (range of spot size is 1.97 to 2.08 mm2 and range of power densities is 66.60 to 73.80 mW). The inclusion of the raw data for these measurements and an estimate of the goodness of fit to the polynomials would better help the reader evaluate whether these parameters are uniform across space and how stable the power density is across repeated stimulations of the same location. Even more helpful would be an estimate of whether the variation in the power density is expected to meaningfully affect the responses of ChR2-expressing sensory neurons.

We thank the reviewer for their comments. As also noted in response to Reviewer 1, the coefficient of variation (CV) is now reported in unitless form (rather than a percentage) to ensure clarity. For avoidance of doubt, the CV is 0.039 (3.9%), so the variation in laser spot size is minimal – there is negligible spot size variability across the system. The ranges are indeed consistent with uniformity. We have included the goodness-of-fit estimates in the appropriate figure legend “fit with a two-dimensional polynomial (area R^2^ = 0.91; power R^2^ = 0.75)”. This indicates that the polynomials fit well overall.

The system already allows for control of spot size. To examine whether the variation in the power density affects the responses of ChR2-expressing sensory neurons, we examined this in our previous work that focused more on input-output relationships, demonstrating a steep relationship between spot size (range of 0.02 mm^2^ to 2.30 mm^2^) and the probability of paw response, demonstrating a meaningful change in response probability (Schorscher-Petcu et al. eLife, 2021). In future studies, we aim to use this approach to “titrate” cutaneous inputs as mice move through their environments.

While the error between the keypoint and laser spot error was reported as ~0.7 to 0.8 mm MAE in Figure 2L, in the methods, the authors report that there is an additional error between predicted keypoints and ground-truth labeling of 1.36 mm MAE during real-time tracking. This suggests that the overall error is not submillimeter, as claimed by the authors, but rather on the order of 1.5 - 2.5 mm, which is considerable given the width of a hind paw is ~5-6 mm and fore paws are even smaller. In my opinion, the claim for submillimeter precision should be softened and the authors should consider that the area of the paw stimulated may differ from trial to trial if, for example, the error is substantial enough that the spot overlaps with the edge of the paw.

We thank the reviewer for identifying a discrepancy in these reported errors. We clarify this below and in the manuscript

The real-time tracking error is the mean absolute Euclidean distance (MAE) between ground truth and DLC on the left hind paw where likelihood was relatively high. More specifically, ground truth was obtained by manual annotation of the left hind paw center. The corresponding DLC keypoint was evaluated in frames with likelihood >0.8 (the stimulation threshold). Across 1,281 frames from five videos of freely exploring mice (30 fps), the MAE was 1.36 mm.

The targeting error is the MAE between ground truth and the laser spot location, so should reflect the real-time tracking error plus errors from targeting the laser. More specifically, this metric was determined by comparing the manually determined ground truth keypoint of the left hind paw and the actual center of the laser spot. Importantly, this metric was calculated using four five-minute high-speed videos recorded at 270 fps of mice freely exploring the open arena (463 frames) and frames were selected with a likelihood threshold >0.8. This allowed us to resolve the brief laser pulses but inadvertently introduced a difference in spatial scaling. After rescaling, the values give a targeting error MAE now in line with the real-time tracking error (see corrected Figure 2L). This is approximately 1.3 mm across all locomotion speeds categories. These errors are small and are limited by the spatial resolution of the cameras. We thank the reviewer for noting this discrepancy and prompting us to get to its root cause.

We have amended the subtitle on Figure 2L as “Ground truth keypoint to laser spot error” and have avoided the use of submillimeter throughout. We have added the following sentence to clarify this point: “As laser targeting relies on real-time tracking to direct the laser to the specified body part, this metric includes any errors introduced by tracking and targeting”.

As the major advance of this paper is the ability to stimulate animals during ongoing movement, it seems that the Figure 3 experiment misses an opportunity to evaluate state-dependent whole-body reactions to nociceptor activation. How does the behavioral response relate to the animal's activity just prior to stimulation?

The reviewers suggest analysis of state-dependent responses. In the Figure 3 experiment, mice were stimulated up to five times when stationary. Analysis of whole body reactions in stationary mice has been described in (Schorscher-Petcu et al. eLife, 2021) and doing this here would be redundant, so instead we now analyse the responses of moving mice in Figure 5. This new analysis shows robust state-dependent responses during movement as suggested by the reviewer. We find two behavioral clusters: one that is for faster, direct (coherent) movement and the other that is for slower assessment (incoherent) movement. Stimulation during the former results in robust and consistent slowing and shift towards assessment, whereas stimulation during the former results in a reduction in assessment. We describe and interpret these new data in the Results and Discussion sections and add information in the Methods and Figure legend, as given below. We believe that demonstrating movement statedependence is a valuable addition to the paper and thank the reviewer for suggesting this.

Given the characterization of full-body responses to activation of TrpV1 sensory neurons in Figure 4 and in the authors' previous work, stimulation of TrpV1 sensory neurons has surprisingly subtle effects as the mice run through the alternating T maze. The authors indicate that the mice are moving quickly and thus that precise targeting is required, but no evidence is shared about the precision of targeting in this context beyond images of four trials. From the characterization in Figure 2, at max speed (reported at 241 +/- 53 mm/s, which is faster than the high speeds in Figure 2), successful targeting occurs less than 50% of the time. Is the initial characterization consistent with the accuracy in this context? To what extent does inaccuracy in targeting contribute to the subtlety of affecting trajectory coherence and speed? Is there a relationship between animal speed and disruption of the trajectory?

We thank the reviewer for pointing out the discrepancy in the reported maximum speed. We have corrected the error in the main text: the average maximum speed is 142 ± 26 mm/s (four mice).

The self-paced T-maze alternation task in Figure 5 demonstrates that mice running in a maze can be stimulated using this method. We did not optimize the particular experimental design to assess the hit accuracy, as this was determined in Figure 2. Instead, we optimized for the pulse frequencies, meaning the galvanometers tracked with processed frames but the laser was triggered whether or not the paw was actually targeted. However, even in this case with the system pulsing in the free-run mode, the laser hit rate was 54 ± 6% (mean ± sem, n = 7 mice). We have weakened references to submillimeter as it was only inferred from other experiments and was not directly measured here. We find in this experiment that stimulation in freely moving mice can cause them to briefly halt and evaluate. In the future, we will use experimental designs to more optimally examine learning.

The reviewer also asks if there is a relationship between speed and disruption of the trajectory. We find that this is the case as described above with our additional analysis.

**Reviewer #3 (Public review)**:Summary:To explore the diverse nature of somatosensation, Parkes et al. established and characterized a system for precise cutaneous stimulation of mice as they walk and run in naturalistic settings. This paper provides a framework for real-time body part tracking and targeted optical stimuli with high precision, ensuring reliable and consistent cutaneous stimulation. It can be adapted in somatosensation labs as a general technique to explore somatosensory stimulation and its impact on behavior, enabling rigorous investigation of behaviors that were previously difficult or impossible to study.Strengths:The authors characterized the closed-loop system to ensure that it is optically precise and can precisely target moving mice. The integration of accurate and consistent optogenetic stimulation of the cutaneous afferents allows systematic investigation of somatosensory subtypes during a variety of naturalistic behaviors. Although this study focused on nociceptors innervating the skin (Trpv1::ChR2 animals), this setup can be extended to other cutaneous sensory neuron subtypes, such as low-threshold mechanoreceptors and pruriceptors. This system can also be adapted for studying more complex behaviors, such as the maze assay and goal-directed movements.Weaknesses:Although the paper has strengths, its weakness is that some behavioral outputs could be analyzed in more detail to reveal different types of responses to painful cutaneous stimuli. For example, paw withdrawals were detected after optogenetically stimulating the paw (Figures 3E and 3F). Animals exhibit different types of responses to painful stimuli on the hind paw in standard pain assays, such as paw lifting, biting, and flicking, each indicating a different level of pain. Improving the behavioral readouts from body part tracking would greatly strengthen this system by providing deeper insights into the role of somatosensation in naturalistic behaviors. Additionally, if the laser spot size could be reduced to a diameter of 2 mm², it would allow the activation of a smaller number of cutaneous afferents, or even a single one, across different skin types in the paw, such as glabrous or hairy skin.

We thank the reviewer for highlighting how our system can be combined with improved readouts of coping behavior to provide deeper insights. Optogenetic and infrared cutaneous stimulation are well established generators of coping behaviors (lifting, flicking, licking, biting, guarding). Detection of these behaviors is an active and evolving field with progress being made regularly (e.g. Jones et al., eLife 2020 [PAWS]; Wotton et al., Mol Pain 2020; Zhang et al., Pain 2022; Oswell et al., bioRxiv 2024 [LUPE]; Barkai et al., Cell Reports Methods 2025 [BAREfoot], along with more general tools like Hsu et al., Nature Communications 2021 [B-SOiD]; Luxem et al., Communications Biology 2022 [VAME]; Weinreb et al,. Nature Methods 2024 [Keypoints-MoSeq]). One output of our system is bodypart keypoints, which are the typical input to many of these tools. We will leave the readers and users of the system to decide which tools are appropriate for their experimental designs - the focus of this current manuscript is describing the novel stimulation approach in moving animals.

**Recommendations for the authors:**
**Reviewer #1 (Recommendations for the authors)**:(1) It is hard to see how the rig is arranged from the render of Figure 2AB due to the components being black on black. A particularly useful part of Fig2AB is the aerial view in panel B that shows the light paths. I suggest adding the labelling of Figure 2A also to that. The side/rear views could perhaps be deleted, allowing the aerial view to be larger.

We appreciate this suggestion and have revised Figure 2B to improve the visibility of the optomechanical components. We have enlarged the side and aerial views, removed the rear view, and added further labels to the aerial view.

(2) MAE - to interpret the 0.54 result, it would be useful to state the arena size in this paragraph.

Thank you. We have added the arena size in this paragraph and also added scales in the relevant figure (Figure 2).

(3) "pairwise correlations of R = 0.999 along both x- and y-axes". Is this correlation between hindpaw keypoint and galvo coordinates?

Yes, we have added the following to clarify: “...between galvanometer coordinates and hind paw keypoints”

(4) Latency was 84 ms. Is this mainly/entirely the delay between DLC receiving the camera image and outputting key point coordinates?

Yes, we hope that the additional detail in the Methods and Discussion described above will now clarify the current closed-loop latencies.

(5) "Mice move at variable speeds": in this sentence, spell out when "speed" refers to mouse and when it refers to hindpaw. Similarly, Fig 2i. The sentence is potentially confusing to general readers (paws stationary although the mouse is moving). Presumably, it's due to gait. I suggest explaining this here.

The speed values that relate to the mouse body and paws are now clearer in the main text and in the legend for Figure 2I.

(6) Figure 2k and associated main text. It is not clear what "success/hit rate" means here.

We have added the following sentence in the main text: “Hit accuracy refers to the percentage of trials in which the laser successfully targeted (‘hit’) the intended hind paw.” and use hit accuracy throughout instead of success rate.

(7) Figure 2L. All these points are greater than the "average" 0.54 reported in the text. How is this possible?

The MAE of 0.54 mm refers to the “predicted and actual laser spot locations” (that is, the difference between where the calibration map should place the laser spot and where it actually fell), while Figure 2L MAE values refers to the error between the ground truth keypoint to laser spot (that is, the error between the human-observed paw target and where the laser spot fell). The latter error will include the former error so is expected to be larger. We have clarified this point throughout the text, for example, stating “As laser targeting relies on real-time tracking to direct the laser to the specified body part, this metric inherently accounts for any errors introduced by the tracking and targeting.”. This is also discussed above in response to Reviewer 2.

(8) "large circular arena". State the size here

We have added this to the Figure 2 legend.

(9) Figure 3c-left. Can the contrast between the mouse and floor be increased here?

We have improved the contrast in this image.

(10) Figure 5c. It is unclear what C1, C2, etc refers to. Mice?

Yes, these refer to mice. We have removed reference to these now as they are not needed.

(11) Discussion. A comment. There is scope for elaborating on the potential for new research by combining it with new methods for measurements of neural activity in freely moving animals in the somatosensory system.

Thank you. We agree and have added more detail on this in the discussion stating “The system may be combined with existing tools to record neural activity in freely-moving mice, such as fiber photometry, miniscopes, or large-scale electrophysiology, and manipulations of this neural activity, such as optogenetics and chemogenetics. This can allow mechanistic dissection of cell and circuit biology in the context of naturalistic behaviors.”

**Reviewer #3 (Recommendations for the authors)**:(1) Include the number of animals for behavior assays for the panels (e.g., Figures 4G).

Where missing, we now state the number of animals in panels.

(2) If representative responses are shown, such as in Figures 3E and 4F, include the average response with standard deviation so readers can appreciate the variation in the responses.

We appreciate the suggestion to show variability in the responses. We have made several changes to Figures 3 and 4. Specifically, to illustrate the variability across multiple trials more clearly, Figure 3E now shows representative keypoint traces for each body part from two mice during their 5 trials. For Figure 4, we have re-analyzed the thermal stimulation trials and shown a raster plot of keypoint-based local motion energy (Figure 4E) sorted by response latency for hundreds of trials. Figure 4G now presents the cumulative distribution for all trials and animals for thermal (18 wild-type mice, 315 trials) and optogenetic stimulation trials (9 Trpv1::ChR2 mice, 181 trials). We also now provide means ± SD for the key metrics for optogenetic and thermal stimulation trials in Figure 4 in the Results section. This keeps the manuscript focused on the methodological advances while showing the trial variability.

(3) "optical targeting of freely-moving mice in a large environments" should be "optical targeting of freely-moving mice in a large environment".

Corrected

(4) Define fps when you first mention this in the manuscript.

Added

(5) Data needs to be shown for the claim "Mice concurrently turned their heads toward the stimulus location while repositioning their bodies away from it".

We state this observation to qualify that the stimulation of stationary mice resulted in behavioral responses “consistent with previous studies”. It would be redundant to repeat our full analysis and might distract from the novelty of the current manuscript. We have restricted this sentence to make it clearer: “Consistent with previous studies, we observed the whole-body behaviors like head orienting concurrent with local withdrawal (Browne et al., Cell Reports 2017; Blivis et al., eLife, 2017.)”